# Gel-assisted mass spectrometry imaging enables sub-micrometer spatial lipidomics

Yat Ho Chan [1], Koralege C. Pathmasiri [1], Dominick Pierre-Jacques [1], Maddison C. Hibbard [1], Nannan Tao[2], Joshua L. Fischer[2], Ethan Yang[2], Stephanie M. Cologna [1,3] & Ruixuan Gao [1,3,4] ✉

A technique capable of label-free detection, mass spectrometry imaging (MSI) is a powerful tool for spatial investigation of native biomolecules in intact specimens. However, MSI has often been precluded from single-cell applications due to the spatial resolution limit set forth by the physical and instrumental constraints of the method. By taking advantage of the reversible interaction between the analytes and a superabsorbent hydrogel, we have developed a sample preparation and imaging workflow named Gel-Assisted Mass Spectrometry Imaging (GAMSI) to overcome the spatial resolution limits of modern mass spectrometers. With GAMSI, we show that the spatial resolution of MALDI-MSI can be enhanced ~3-6-fold to the sub-micrometer level without changing the existing mass spectrometry hardware or analysis pipeline. This approach will vastly enhance the accessibility of MSI-based spatial analysis at the cellular scale.

With recent advances in soft ionization techniques, such as matrix-assisted laser desorption/ionization (MALDI) and desorption electrospray ionization (DESI), mass spectrometry (MS)-based spatial analysis (mass spectrometry imaging or MSI) has drastically pushed the methodological boundaries of spatial biology research[1–3]. MSI exhibits an inherent capability for label-free detection, which grants it distinct advantages over other imaging modalities in spatial molecular profiling of native biomolecules, such as lipids, proteins, metabolites, and glycans. In particular, high spatial resolution MSI has great utility in uncovering the molecular constituents driving cellular functions and dysfunctions in intact biological specimens[4]. For example, (sub)cellular MALDI-MSI studies can be used to identify native lipids and proteins associated with metabolic homeostasis, membrane potential regulation, and protein misfolding/aggregation within the original tissue microenvironments[5,6].

To date, the spatial resolution of MSI has been limited by both physical and instrumental constraints. For example, the spatial resolution of MALDI-MSI is defined by the matrix crystal size, which is typically around a few to 10 micrometers depending on the matrix type and application method[7,8], as well as the instrument operating pixel size, which generally falls between 5 to 10 s of micrometers as dictated by the laser spot size and stage precision. A handful of advanced MSI techniques, including secondary ion mass spectrometry (SIMS)[9–11], custom-built MALDI-TOF mass spectrometers[12,13], and MALDI-2-enhanced sample imaging[14–16], have demonstrated (sub)cellular-level spatial analysis of intact specimens. However, these methods are often not used on a daily basis in the existing biomedical labs due to their limited accessibility or mass range (m/z, mass-to-charge ratio) of ions that can be detected. Such fundamental and practical limitations have slowed down the application of MSI to single-cell studies, including spatial lipidomics and proteomics.

Pioneering studies in MSI and physical stretching of biological specimens have provided valuable clues to overcome these limitations. For example, previous studies have demonstrated single-cell-sized tissue fragmentation and more than an order of magnitude increase in pixel density[17,18]. The recent introduction of expansion microscopy has enabled a more uniform sample stretching by (bio) chemically breaking down gelled specimens[19]. In expansion microscopy, biomolecules of interest are retained within a swellable hydrogel before the sample-hydrogel composite is allowed to break down at the molecular level by proteolysis or similar treatment to subsequently expand several folds. The resulting resolution-enhancing

[1]Department of Chemistry, University of Illinois Chicago, Chicago, IL, USA. [2]Bruker Daltonics, Billerica, MA, USA. [3]Laboratory for Integrative Neuroscience, University of Illinois Chicago, Chicago, IL, USA. [4]Department of Biological Sciences, University of Illinois Chicago, Chicago, IL, USA. ✉e-mail: gaor@uic.edu

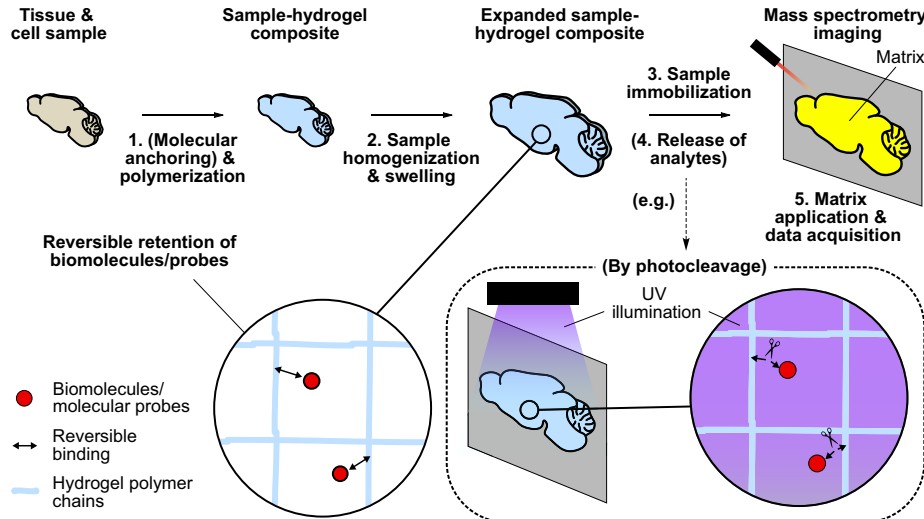

**Fig. 1 | Gel-assisted mass spectrometry imaging (GAMSI).** Schematics showing the principle and generalized workflow of GAMSI. Samples are (**1**) treated (optionally) with cleavable small-molecule linkers and polymerized to form a superabsorbent hydrogel composite, (**2**) homogenized and expanded, (**3**) immobilized onto a sample plate, (**4**) subject (optionally) to cleavage of the small-molecule linkers to release the analytes via, for example, a photocleavage event (dotted box), and (**5**) coated with a matrix (if used) and analyzed on a mass spectrometer. The magnified schematics (solid circles) show the reversible tethering/release of biomolecules and molecular probes to/from the hydrogel polymer chains in the sample-hydrogel composite.

and decrowding effects of expansion microscopy have made it a powerful method for super-resolution fluorescence microscopy and spatial multi-omics[20–22]. Recent studies have shown that the proteomic profile of tissue samples can be determined by manually dissecting an expanded sample and analyzing each part with existing LC-MS instruments[23,24]. A different study combining expansion microscopy and multiplexed ion beam imaging (MIBI) has demonstrated the compatibility of the highly-multiplexed, isotope-conjugated antibody labeling and mass spectrometry-based protein identification[25].

These early studies and applications have paved the way for our conceptualization of direct MSI of native biomolecules, including lipids and proteins, in a hydrogel-embedded sample. We hypothesized that target biomolecules can be readily desorbed and ionized for mass spectrometry analysis if their (or their labels') interactions with the hydrogel polymer network are made more reversible. Based on this concept, we have developed a sample preparation and imaging method that overcomes the spatial resolution limit of existing mass spectrometers. We name this method Gel-Assisted MSI (or GAMSI), as biological specimens are polymerized (and expanded) using a superabsorbent hydrogel to assist and enhance the conventional MSI capability (Fig. 1). For the initial demonstration, we have focused on MALDI-MSI's relative quantification capability because it is widely used by the field and does not require the introduction of additional standards or calibrations. With GAMSI, we show that the spatial resolution of lipid and protein MALDI-MSI can be enhanced ~3-6-fold without modifying the hardware or the data analysis pipeline.

## Results

### Principle and general workflow of Gel-Assisted Mass Spectrometry Imaging (GAMSI)

For a general GAMSI workflow, we propose using a molecular retention strategy that can reversibly tether and release the analytes to and from the hydrogel polymer network (Fig. 1). As opposed to the irreversible anchoring scheme adopted by conventional expansion methods, this approach would allow in situ desorption and detection of biomolecules or probes in the subsequent MSI measurements.

We note that this reversible anchoring can take place non-covalently or covalently. For lipid GAMSI, for example, native lipids can be tethered to the hydrogel via hydrophobic interactions with the membrane proteins that are covalently anchored to the hydrogel polymer chains. These lipids can then be desorbed and ionized through interactions with the MALDI matrix and organic solvents prior to MALDI-MSI. To verify if such non-covalent retention of lipids is possible, we assessed the overall retention rate of lipids in the expanded tissue with or without surfactant treatment, which is known to disrupt the hydrophobic interactions between the lipids and membrane proteins (Supplementary Fig. 1). We found that the surfactant-free controls drastically increased the lipid retention compared to the surfactant-treated samples. This suggests that a large portion of native lipids were retained in the surfactant-free samples through hydrophobic interactions. Moreover, recent biophysical and computational studies have found that the membrane proteins' hydrophobic interactions with lipids can extend as far as a few nanometers in the membrane[26,27]. While the actual radius of lipids tethered to the membrane proteins remains to be studied for GAMSI, these results suggest that lipids can indeed be retained around the anchored membrane proteins via hydrophobic interactions.

For GAMSI, the target biomolecules (or their labels) can also be covalently tethered to the hydrogel polymer chains using a cleavable linker. Once anchored, these molecules and labels can be released from the hydrogel by cleaving the linker. For protein GAMSI, for example, targeted proteins can be labeled with an antibody carrying a photocleavable mass reporter[28]. These antibodies bearing the mass reporters can then be covalently anchored to the hydrogel using a small-molecular linker that binds proteins to the hydrogel polymer chains (e.g., Acryloyl-X, SE)[29,30]. After sample expansion and immobilization, the mass reporters conjugated to the target proteins can be released by a photocleavage step.

### GAMSI sample preparation and retention of native biomolecules

To validate the working principle of GAMSI, we first confirmed that native lipids could be efficiently retained and imaged after gel-assisted sample expansion. To enhance lipid retention, we optimized the GAMSI sample preparation workflow, including the fixation and homogenization steps. Fresh-frozen tissue is one of the most commonly used sample formats for MALDI-MSI. For GAMSI, however, we found that a light chemical fixation with paraformaldehyde (PFA) or a combination of PFA and glutaraldehyde (GA) (FPA/GA) was a better

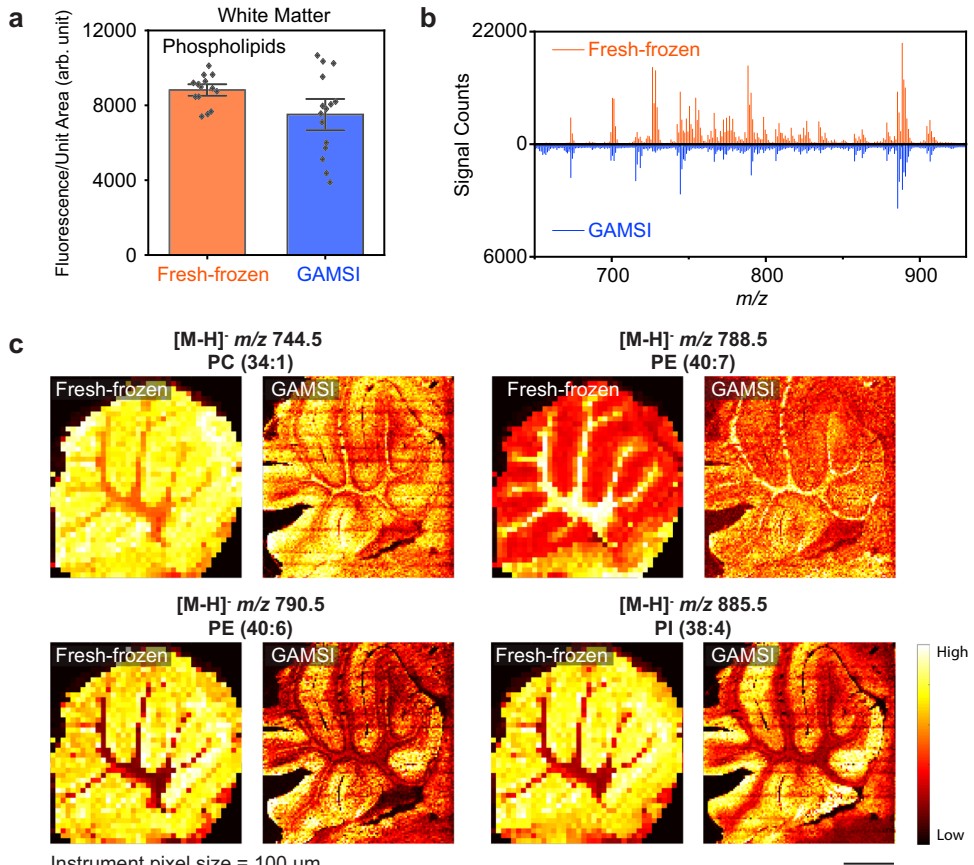

**Fig. 2 | Lipid GAMSI. a** Fluorescence intensity per normalized unit tissue area of fresh-frozen and GAMSI-processed (PFA/GA-fixed) mouse cerebellum white matter, fluorescently labeled using a phospholipid dye [bar height, mean; black dots, individual data points; error bar, standard error of the mean (SEM); $n$ = 15 regions of interest (ROIs) from three brain slices from one animal]. The unit tissue area was normalized to the pre-expansion scale. **b** Averaged mass spectra ($m/z$ = 650-925) of fresh-frozen and GAMSI-processed (PFA/GA-fixed) mouse cerebellum. **c** Spatial distributions of selected lipids in the fresh-frozen (left) and GAMSI-processed (PFA/GA-fixed, ~3-fold expanded) (right) mouse cerebellum. The instrument pixel size (raster distance) (AB SCIEX 4800) was set at 100 μm. Scale bar: 1 mm (3 mm).

Here and after, unless otherwise noted, color scale bars for mass spectrometry images represent the relative intensity of the signals, $m/z$ (mass-to-charge ratio) values are provided for singly charged deprotonated ions [M-H]⁻, and scale bars are provided at the pre-expansion scale (with the corresponding post-expansion size indicated in the brackets). The instrument pixel size refers to the physical size of a pixel as determined by the instrument specifications, whereas the effective pixel size is given as the instrument pixel size divided by the sample expansion factor. PC: phosphatidylcholine; PE: phosphatidylethanolamine; PI: phosphatidylinositol. Source data are provided as a Source Data file.

choice because it enhanced the sample integrity for the subsequent polymerization step while maintaining a lipid profile comparable to that of the fresh-frozen sample (Supplementary Fig. 2)[31,32]. Hence, unless otherwise noted, all the GAMSI samples were pre-processed using this light fixation step.

In the context of gel-assisted expansion, sample homogenization is a crucial step that (bio)chemically breaks down the structural integrity of the gelled specimen while preserving its original spatial and molecular information to a large extent. In the original expansion protocol, proteinase K (proK)-based proteolysis and high-concentration surfactant treatment were used to homogenize the sample-hydrogel composite so that it expands more isotropically[19,29,30]. However, during this process, most (if not all) native lipids are lost due to the surfactant treatment and the strong digestion (Supplementary Fig. 1). To circumvent this issue, we reasoned that trypsin-based proteolysis with reduced or no surfactants should be sufficient to homogenize the thin specimens (~5–30 μm) commonly used for MALDI-MSI studies. To validate this, we treated up to ~40 μm thick mouse brain slices with a surfactant-free trypsin digestion buffer (Supplementary Fig. 3). After homogenizing and expanding the tissue, we observed no visible sample breaking or tearing, which indicated that the sample homogenization was

complete and sufficient. To further evaluate the effectiveness of this homogenization step, we characterized the global expansion isotropy of mouse brain slices (~25 μm) treated with the same surfactant-free trypsin digestion by comparing the same tissue features before and after the expansion (Supplementary Fig. 4). We found that the average expansion error of ~1–4% of the trypsin-digested tissue slices was comparable to the ~1-5% benchmark reported for proK-digested tissue slices[19,29]. Combined, these results show that trypsin-based proteolysis is sufficient to homogenize thin tissue samples for the subsequent expansion and imaging. We note that an additional benefit of switching from proK to trypsin digestion is that the latter would provide a more predictable proteolytic profile, which will be critical for untargeted proteomic analysis in future studies.

To estimate the percentage of native lipids retained after gel-assisted expansion, we fluorescently labeled fresh-frozen (as control) and GAMSI-processed mouse cerebellum slices with a phospholipid dye and measured the fluorescence intensity per normalized unit tissue area (Fig. 2a, Supplementary Fig. 5). We found that ~84% of the phospholipid labeling was preserved in the white matter after the gel-assisted expansion. In addition, ~82% and ~100% of the phospholipid labeling was preserved in the molecular layer and the granular layer,

respectively. These results show that, regardless of the tissue region and lipid abundance, a majority of the native lipids can be preserved after sample homogenization and expansion.

Lastly, we obtained averaged mass spectra of both the fresh-frozen and GAMSI-processed mouse cerebellum slices on a MALDI-TOF mass spectrometer (Applied Biosystems, "AB SCIEX 4800") (Fig. 2b). We found that the GAMSI-processed sample replicated most of the major lipid peaks (in the 650-925 $m/z$ range) in the fresh-frozen sample. To better assess the difference in their lipid profile, we tabulated lipid peaks found commonly across the two samples and those only detected in one of the samples (Supplementary Fig. 6a). We found that of all the lipid peaks observed in the fresh-frozen sample, ~49% were also detected in the GAMSI sample. We note that this percentage can be further increased by utilizing a mass spectrometer of higher sensitivity (see later discussion). To evaluate chemical perturbations introduced by GAMSI, we further analyzed the chemical breakdown of the detected lipid peaks (Supplementary Fig. 6b). We found that the lipid composition stayed largely constant across the fresh-frozen, the common, and the GAMSI lipid peaks. This suggests that GAMSI did not substantially alter or bias the retained lipids toward a specific chemical composition. While a more comprehensive mass spectrometry characterization (e.g., using on-sample tandem MS and bulk LC-MS) is necessary for future studies to fully evaluate the change in the lipidomic profile, these results provide a preliminary, yet crucial assessment of the chemical perturbation introduced by the GAMSI sample preparation.

### Enhancing the spatial resolution of lipid mass spectrometry imaging

Next, to verify the spatial resolution enhancing capability, we performed lipid GAMSI on a mouse cerebellum sagittal slice and collected the same imaging data on a fresh-frozen counterpart as the control (Fig. 2c). As a result, we observed an increase in the imaging resolution that corresponded to the ~3-fold sample expansion. In addition, we observed a similar spatial distribution for the major lipid peaks across the two samples (Supplementary Figs. 7–9). It is important to note that the observed change in the spatial resolution corresponded to nearly an order of magnitude increase in the pixel density, as the number of pixels per normalized unit tissue area scales with the square of the expansion factor (~9-fold = ~3 × 3). In addition, to validate the identity of the detected lipids, we performed on-tissue tandem MS on several lipid peaks from the GAMSI sample (Supplementary Fig. 10). We confirmed that the fragmentation patterns of these lipid peaks were consistent with those reported in the previous tandem MS studies[33–35]. These results show that GAMSI can indeed preserve and replicate both the spatial and molecular information of the fresh-frozen samples.

In principle, the concept and workflow of GAMSI are applicable to all MSI pipelines. Therefore, an increase in spatial resolution should be observed on any MALDI mass spectrometer using GAMSI. To validate this, we performed lipid GAMSI on a Bruker rapifleX MALDI Tissuetyper ("Bruker rapifleX"). Similar to the AB SCIEX 4800 results, the expanded mouse cerebellum images captured the native molecular and spatial information of the fresh-frozen counterpart to a large extent (Fig. 3, Supplementary Fig. 11). Moreover, a ~4-fold enhancement of spatial resolution similarly corresponded to the observed expansion factor of the sample. These results suggest that GAMSI can increase the spatial resolution of MALDI-MSI independent of the mass spectrometer hardware. As with the AB SCIEX 4800 results, we further analyzed the lipid coverage in the fresh-frozen and GAMSI mass spectra and found that significantly more lipids were detected for both the fresh-frozen and GAMSI sample (Supplementary Fig. 12a). In addition, we observed that a higher percentage (~57%) of fresh-frozen lipid peaks were co-detected by GAMSI, suggesting that mass spectrometers with higher sensitivity may aid detection of more fresh-frozen lipid peaks in GAMSI. Lastly, consistent with the AB SCIEX 4800 results, the chemical composition of lipids remained largely unchanged across the fresh-frozen, the common, and the GAMSI lipid peaks (Supplementary Fig. 12b).

### Enhancing the spatial resolution of multiplexed lipid-protein mass spectrometry imaging

The capability to identify and spatially map multiple types of biomolecules makes MSI a powerful tool for spatial multi-omics studies. Hence, we tested if GAMSI could enhance the spatial resolution of multiplexed MSI. As a proof-of-concept, we introduced an additional protein labeling step after lipid GAMSI by using photocleavable mass-tag conjugated antibodies[28] (Supplementary Fig. 13). We chose targeted protein imaging for our initial demonstration because the molecular specificity and spatial distribution of antibody labeling have been validated by other methods such as western blot and immunohistochemistry.

Figure 4 shows the spatial distribution of the imaged proteins and selected native lipids from ~4-fold expanded mouse cerebellum. The spatial arrangements of myelin basic protein (MBP) and Synapsin I (SYN-I) matched those validated by immunofluorescence and conventional MSI[28]. We note that this lipid-protein GAMSI workflow can be potentially scaled up to tens or more protein targets since the mass-tag labeling is not limited by the spectral overlap that often hinders molecular multiplexing using fluorescence probes[36]. However, for multiplexed imaging, it is critical to test the staining quality of the conjugated antibodies and switch to validated ones, when necessary, as we have observed low signal counts and aggregations with some of the existing probes. In future studies, lipid-protein GAMSI can be further extended to untargeted protein imaging, given the recent success in combining mass-tag and untargeted proteomic analyses[37].

### Sub-micrometer mass spectrometry imaging using off-the-shelf MALDI-TOF mass spectrometers

The multifold improvement in the spatial resolution offered by GAMSI opens up the possibility of detecting cellular molecular features with conventional MALDI-MSI instruments. For example, with ~4-fold expansion, we were able to capture the enrichment of specific lipids in the Purkinje cells at single-cell resolution (Fig. 5a). Given the ever-improving detection sensitivity of modern mass spectrometers, we reasoned that the effective spatial resolution of GAMSI could be further enhanced by decreasing the instrument pixel size and increasing the sample expansion factor. To demonstrate this, we performed lipid GAMSI using a Bruker timsTOF fleX ("Bruker timsTOF") with ~4-fold and ~6-fold expanded mouse cerebellum (Fig. 5b, c). Extending upon our previously analyzed ~4-fold expanded mouse cerebellum slices, the instrument pixel size was first decreased from 20 μm to 5 μm on the Bruker timsTOF (Fig. 5b). We found that we could obtain reasonably high signal counts per pixel by adjusting the imaging conditions (e.g., the number of laser shots per pixel) as the corresponding effective pixel size was reduced from ~5 μm to ~1.3 μm (obtained by dividing the instrument pixel size by the sample expansion factor).

Next, we increased the sample expansion factor from ~4-fold to ~6-fold while keeping the instrument pixel size constant at 5 μm. We confirmed that the ~6-fold expanded sample imaged on the Bruker timsTOF replicated major lipid peaks (in the 500-1000 $m/z$ range in negative ion mode) of the ~4-fold expanded sample imaged on the Bruker rapifleX (Supplementary Fig. 14). Importantly, the signal counts for the ~6-fold expanded sample stayed sufficiently high after adjusting the imaging parameters. As a result, we were able to achieve lipid GAMSI in intact tissue with sub-micrometer effective pixel size (~880 nm) (Fig. 5c). With this effective spatial resolution, we were able to identify, for example, the glial and neuronal nuclei (~4-6 μm in diameter) in the cerebellar white matter. Combined, these results show that GAMSI can indeed achieve higher spatial resolution by decreasing the instrument pixel size or increasing the sample expansion factor.

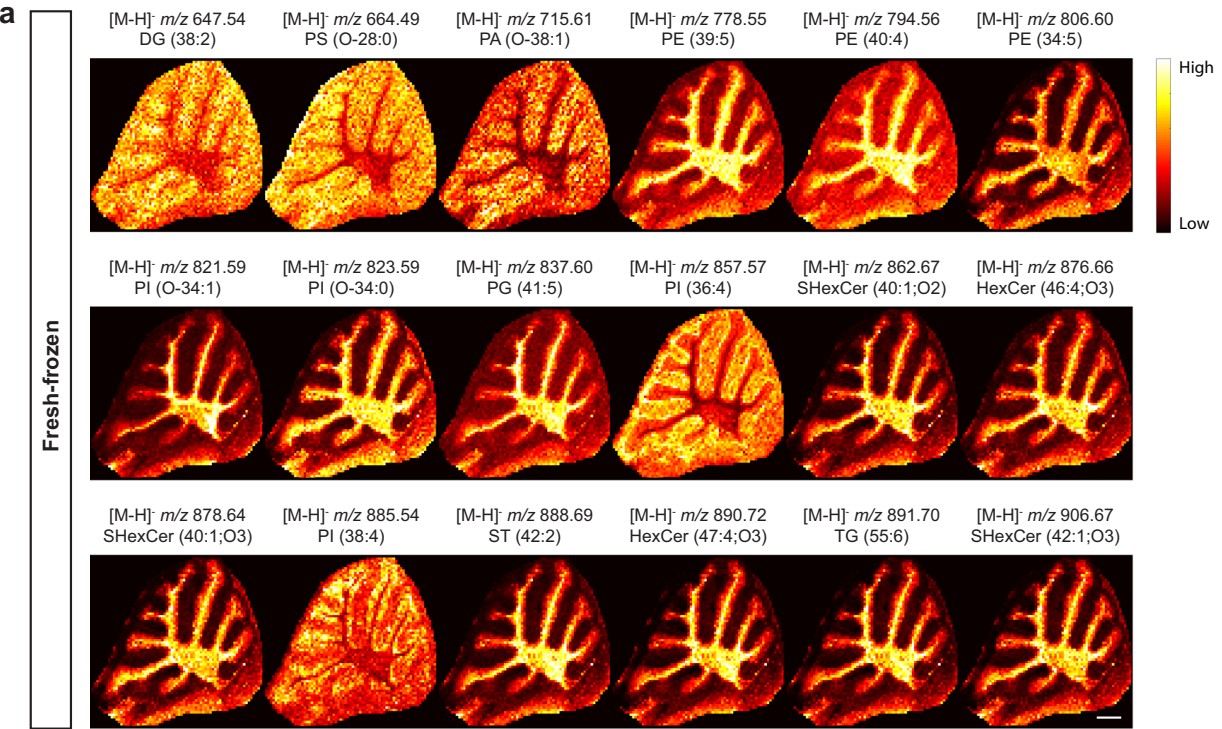

Instrument pixel size = 50 μm

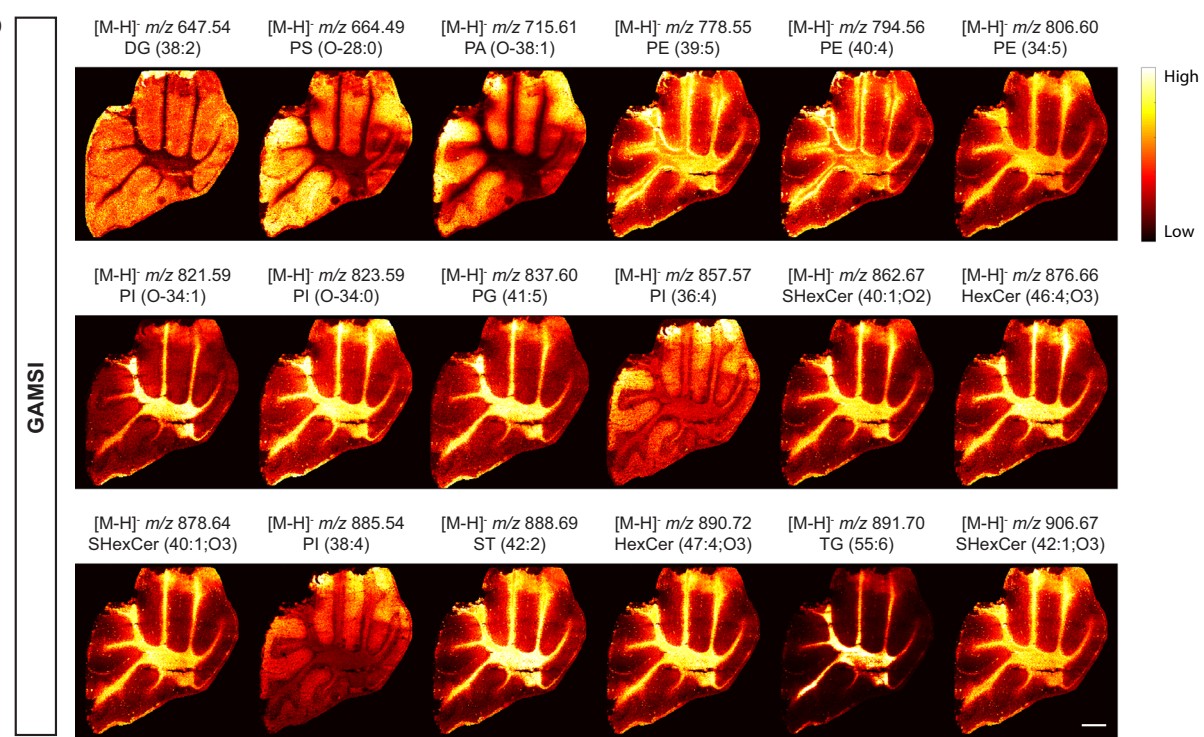

Instrument pixel size = 50 μm

**Fig. 3 | GAMSI is applicable to any MSI pipelines.** Spatial distributions of selected lipids in (**a**) fresh-frozen and (**b**) GAMSI-processed (PFA-fixed, ~4-fold expanded) mouse cerebellum. Both samples were imaged with the instrument pixel size (Bruker rapifleX, with 9AA as the default matrix for lipid imaging) set at 50 μm.

Scale bars: 500 μm (1.9 mm). DG: diacylglycerol; PS: phosphatidylserine; PA: phosphatidic acid; PE: phosphatidylethanolamine; PG: phosphatidylglycerol; PI: phosphatidylinositol; SHexCer: sulfatide hexosylceramide; HexCer: hexosylceramide; ST: sterol; TG: triglyceride.

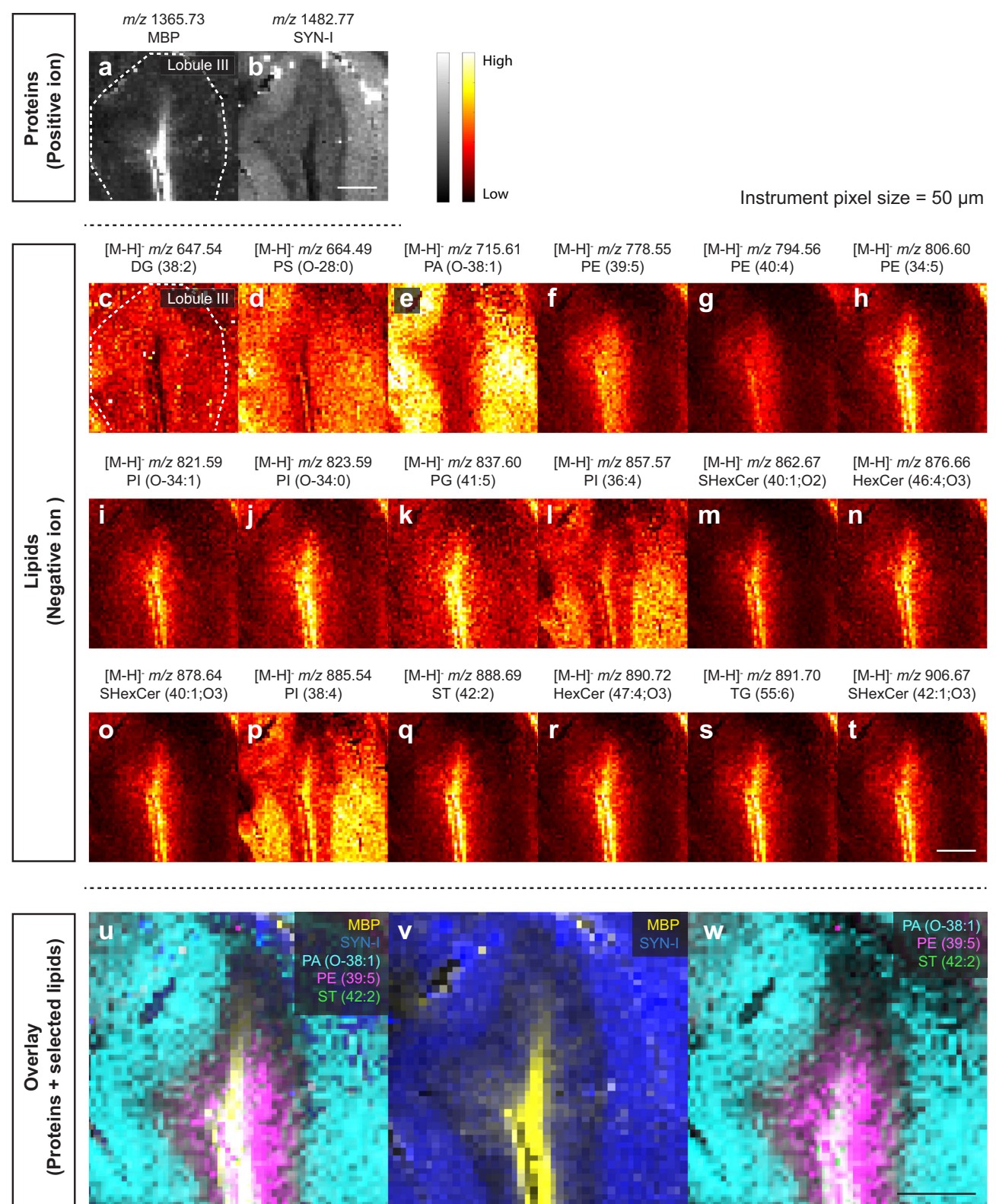

**Fig. 4 | Multiplexed lipid-protein GAMSI.** Spatial distributions of (**a**–**b**) mass reporters of photocleavable mass-tags (PC-MTs) targeting MBP and SYN-I, and (**c**–**t**) selected lipids in a - 4-fold expanded mouse cerebellum lobule (PFA-fixed). **u**–**w** Overlays of proteins and selected lipids. The instrument (Bruker rapifleX) pixel size was set at 50 μm. Scale bars: 200 μm (760 μm). MBP: myelin basic protein; SYN-I: Synapsin I; DG: diacylglycerol; PS: phosphatidylserine; PA: phosphatidic acid; PE: phosphatidylethanolamine; PG: phosphatidylglycerol; PI: phosphatidylinositol; SHexCer: sulfatide hexosylceramide; HexCer: hexosylceramide; ST: sterol; TG: triglyceride.

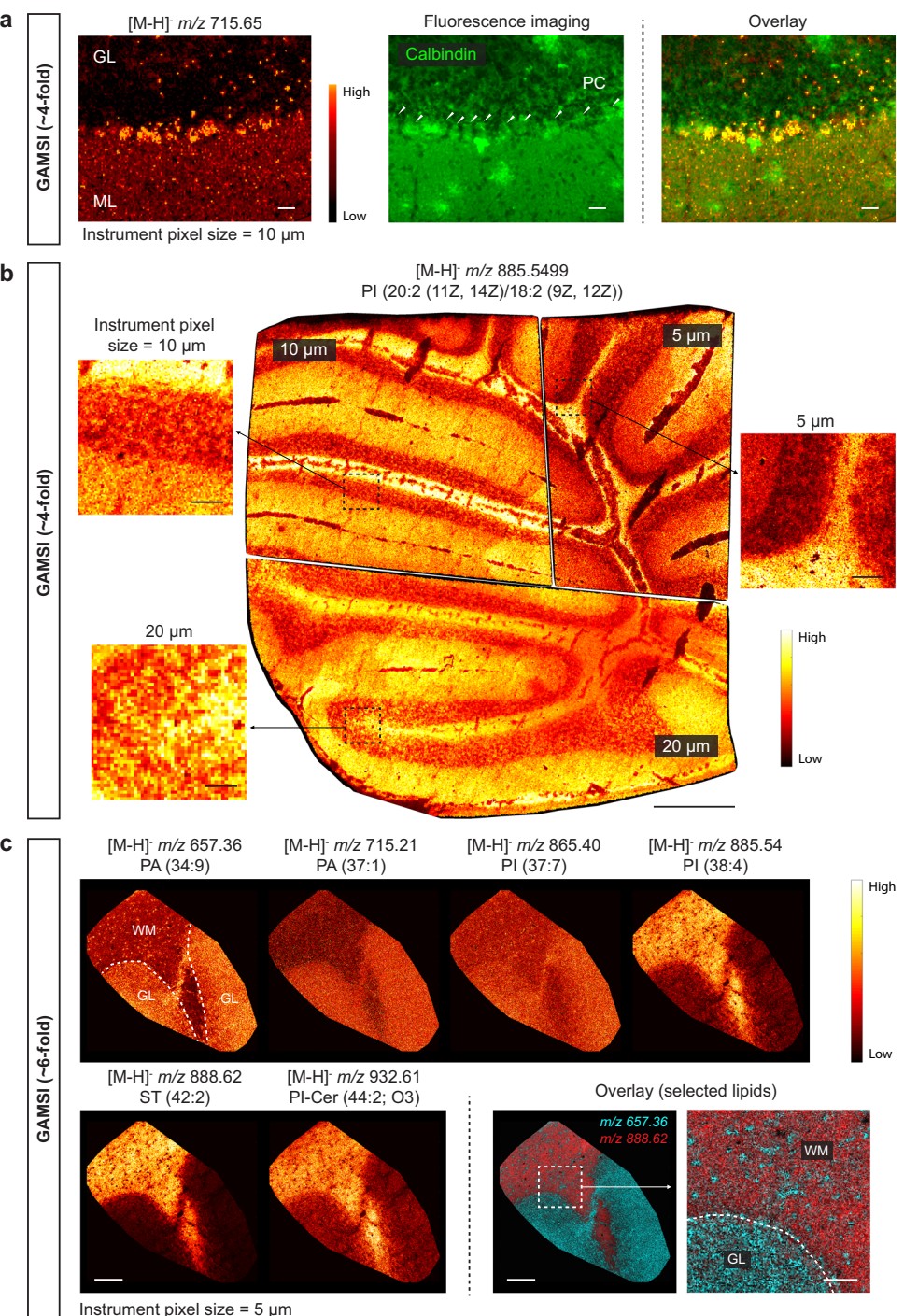

**Fig. 5 | Lipid GAMSI at sub-micrometer resolution. a** Lipid GAMSI of ~4-fold expanded mouse cerebellum (PFA-fixed) with the instrument (Bruker rapifleX, with DAN as the matrix) pixel size set at 10 μm (left). The effective pixel size is ~2.5 μm. Fluorescence image of the same tissue slice (immunostained against calbindin) and an overlay of the GAMSI and fluorescence images are shown to the right. The arrows in the fluorescence image indicate individual Purkinje cells. Scale bars: 25 μm (100 μm). The experiment was repeated twice. **b** Lipid GAMSI of ~4-fold expanded mouse cerebellum (PFA-fixed) with the instrument (Bruker timsTOF) pixel size set at 20 μm (bottom), 10 μm (top left), and 5 μm (top right). The corresponding effective pixel size is ~5 μm, ~2.5 μm and ~1.3 μm, respectively. Scale bar:

500 μm (1.9 mm). (Insets) Magnified views of the boxed regions. Scale bars: 50 μm (190 μm). The apparent tissue tears are from cryo-sectioning. Unless otherwise noted, all the Bruker timsTOF images were collected using MALDI-1 mode. **c** Lipid GAMSI of ~6-fold expanded mouse cerebellum (PFA-fixed) with the instrument (Bruker timsTOF) pixel size set at 5 μm. The effective pixel size is ~880 nm. Scale bars: 100 μm (570 μm), and 30 μm (171 μm) for the magnified view of the boxed region in the overlay image. The apparent tissue tears are from cryo-sectioning. The experiment was repeated once. WM: white matter; GL: granular layer; ML: molecular layer; PC: Purkinje cell; PI: phosphatidylinositol; PA: phosphatidic acid; ST: sterol; PI-Cer: ceramide phosphoinositol.

## Discussion

Currently, the spatial resolution of MALDI-MSI is limited by both physical and instrumental constraints of the method and is often too coarse for single-cell studies. To overcome these limitations, we have proposed to take advantage of reversible interactions between the analytes and a superabsorbent hydrogel, and established a sample preparation and imaging method named GAMSI. Our results show the spatial resolution of lipid MALDI-MSI can be enhanced by gel-assisted sample expansion without the need to modify the existing MSI hardware or analysis pipeline. Our study further suggests that such an approach can be extended to multiplexed imaging of multiple types of molecular targets (such as lipids and proteins) and to sub-micrometer scale imaging by tuning the expansion factor.

Due to gel-assisted expansion, the density of target biomolecules and labels in the sample is diluted in proportion to the square of its expansion factor. Given this dilution effect, the instrument detection sensitivity becomes a limiting factor for the achievable effective spatial resolution in GAMSI. One solution to overcome this limitation is to perform MSI on a mass spectrometer of higher sensitivity. For MALDI-MSI, switching to post-ionization (e.g., MALDI-2) is another potential solution to improve the sensitivity[14–16]. Other factors associated with the sample geometry and preparation may also help remedy this dilution effect. For example, the number of analytes per imaging area can be increased by increasing the thickness of the tissue slice as the expanded sample is collapsed to a planar geometry Here, more analytes per post-expansion unit area may be accessed by the matrix.

Another limiting factor for GAMSI is throughput. If the time spent on each pixel remains constant, the total imaging time increases drastically for the GAMSI sample as the imaging area increases proportionally to the square of the expansion factor. In practice, GAMSI imaging sessions often take multiple hours to overnight on the existing MALDI-TOF mass spectrometers. A potential solution is to, for example, decrease the number of laser shots per pixel. However, this number also needs to stay above a certain value to maintain a reasonable signal-to-noise ratio. Additional optimization is thus necessary to find the appropriate number of laser shots per pixel for each instrument and sample type.

To summarize, our approach will accelerate the daily usage of MSI for single-cell profiling of native biomolecules, especially for sub-micrometer spatial lipidomic studies of intact cells and tissues. As discussed, dilution of analytes and dilation of imaging time are the major trade-offs for the enhanced spatial resolution. However, as the sensitivity and imaging speed of modern mass spectrometers continue to improve with the ongoing innovations in the field, even higher enhancement of spatial resolution will be possible using this approach.

## Methods

### Ethics statement

The animal care and handling procedures, including basic care, housing and husbandry, and surgical procedures, were performed in accordance with the US National Institutes of Health Guide for the Care and Use of Laboratory Animals and approved by the University of Illinois Chicago Animal Care Committee. Sex and gender were not considered due to the proof-of-concept nature of this study.

### Chemicals and reagents

All reagents were used as supplied unless otherwise noted. Purified water was obtained from a Milli-Q IQ 7000 Ultrapure Water System (Millipore Sigma). All chemicals and reagents were obtained from Millipore Sigma unless otherwise noted.

### Animals

Adult, both male, and female, heterozygous BALB/cNctr-*Npc1*^mlN/J mice (Jackson Laboratory) were maintained in a breeding colony at the University of Illinois Chicago. All animals were housed in appropriate cages with 12-hour dark and 12-hour light cycles, ambient temperature, and humidity.

### Tissue preparation

At 7 weeks of age, the mice were euthanized via $CO_2$ asphyxiation followed by decapitation. Whole brains were dissected and immediately frozen on dry ice to maintain tissue integrity. The brains were stored at −80 °C until sectioning. Brain slices of ~25 μm were collected on a CryoStar NX50 Cryostat (Epredia) and thaw-mounted on positively charged microscope slides (MIDSCI) for GAMSI sample preparation, or on stainless steel MALDI plates (Applied Biosystems SCIEX) or ITO-coated glass slides (PL-IF-000010-P25, Hudson Surface Technology, Inc.) for the fresh-frozen control. The sectioned mouse brain slices were stored at −80 °C until further analysis and procedure.

### Sample gelation

The ~25 μm thick mouse brain slices were thawed to room temperature and fixed with 4% paraformaldehyde (PFA) (047340.9 M, Thermo Fisher) or a combination of 3% PFA and 0.1% glutaraldehyde (GA) (50-262-10, Thermo Fisher) (for lipid imaging only) at room temperature for 10 min. The fixatives were removed, and the brain slices were washed with 1x PBS three times for 5 min each time. Next, the brain slices were incubated in an Acryloyl-X, SE (AcX, A20770, Thermo Fisher) solution (0.1 mg/mL in 1x PBS) at room temperature for 6 hours, and washed with 1x PBS twice for 15 min each time. Gelation of the brain slices was performed following a modified version of the previously described protocols[29,30]. Briefly, for ~3-4-fold expansion, the brain slices were incubated with the monomer solution [1x PBS, 2 M NaCl, 8.625% (w/v) sodium acrylate, 2.5% (w/v) acrylamide, 0.15% (w/v) N,N'-methylenebisacrylamide, 0.01% (w/v) of 4-hydroxy-2,2,6,6-tetramethylpiperidin-1-oxyl (4HT), 0.2% (w/v) of ammonium persulfate (APS), and 0.2% (w/v) of tetramethylethylenediamine (TEMED)] at 4 °C for 30 min and gelled in a humidified 37 °C incubator for 2 hours. For ~6-fold expansion, the brain slices were incubated with the monomer solution [1x PBS, 1.5 M NaCl, 10.34% (w/v) sodium acrylate, 14.2% (w/v) acrylamide, 0.005% (w/v) N,N'-methylenebisacrylamide, 0.0015% (w/v) of 4HT, 0.15% (w/v) of APS, and 0.15% (w/v) of TEMED] at 4 °C for 30 min and gelled in a humidified 37 °C incubator for 2 hours.

### Sample homogenization

The gelled brain slices were homogenized in a trypsin (20233, Thermo Fisher) digestion buffer [0.025-0.25% (w/v) in 1x PBS] at 37 °C for 2-4 days with the trypsin digestion buffer freshly prepared each day.

### Sample expansion and immobilization

The gelled and digested brain slices were placed in an excess volume of 0.5x PBS for 20 min and then in purified water three times for 20 min each time until the samples were fully expanded. Next, the expanded samples were transferred onto the stainless steel MALDI plates (Applied Biosystems SCIEX) or ITO-coated glass slides (PL-IF-000010-P25, Hudson Surface Technology, Inc.) or MALDI IntelliSlides (1868957, Bruker) using a 36 ×60 mm no. 1.5 cover glass (260461-100, Ted Pella). Finally, the samples were dried under vacuum in a Pyrex vacuum desiccator (Millipore Sigma) filled with Drierite (Millipore Sigma) at room temperature for 4 hours to overnight, and then in a CentriVap Benchtop Vacuum Concentrators (Labconco) at 37 °C for 2 minutes to ensure complete removal of moisture.

### Matrix application

For lipid imaging on the 4800 Plus MALDI TOF/TOF Analyzer (Applied Biosystems SCIEX) and rapifleX MALDI Tissuetyper (Bruker), 1,5-diaminonaphthalene (DAN) was applied to the immobilized and dried sample via sublimation using a previously described homemade sublimation apparatus[38,39]. Briefly, 50 mg of DAN was dissolved in 2 mL of acetone, aspirated onto the bottom of the sublimation flask, and blow-

dried using nitrogen to form a thin layer of white solid. Next, a hot-plate, to which the sublimation flask was placed, was set to 105 °C while a digital thermometer was placed in contact with the bottom of the flask to monitor the temperature. An ice slush was added to the cold finger of the apparatus, to which the MALDI plate or the glass slide with the sample was adhered on the underside with copper tape. Finally, the sublimation apparatus was placed under vacuum at 80 mTorr using a rough pump, and the DAN matrix was sublimed for 2 min. The amount of matrix deposited on the sample was determined as mass per centimeter square. For Bruker rapifleX imaging, the sample coated with the matrix was recrystallized at 55 °C for 2 min in a humidified chamber containing 5% isopropyl alcohol (1027811000, Millipore Sigma).

For lipid imaging on the rapifleX MALDI Tissuetyper (Bruker) and timsTOF fleX MALDI-2 (TTF) (Bruker), 9-aminoacridine (9AA) solution was sprayed to the sample with a HTX M3+ sprayer (HTX Technologies, LLC) using the following settings: Nozzle temperature = 60 °C; nozzle velocity = 1200 mm/min; flow rate = 0.12 mL/min; 9AA concentration = 10 mg/mL; number of passes = 8; track spacing = 2 mm; nitrogen gas pressure = 10 psi. The 9AA solution was prepared by dissolving 100 mg of 9AA to a 10 mL solution of 75% ACN and 25% purified water, followed by filtering.

For PC-MT imaging on rapifleX MALDI Tissuetyper (Bruker), α-cyano-4-hydroxycinnamic acid (CHCA) solution [7 mg/mL in 50% ACN, 50% purified water, and 0.1% trifluoroacetic acid (TFA)] was sprayed to the sample with the HTM M3+ sprayer using the following settings: Nozzle temperature = 79 °C; nozzle velocity = 1200 mm/min; flow rate = 0.1 mL/min; CHCA concentration = 7 mg/mL; number of passes = 10; track spacing = 2 mm; nitrogen gas pressure = 10 psi. Prior to imaging, the sample coated with the matrix was recrystallized at 55 °C for 2 min in a humidified chamber containing 5% isopropyl alcohol (1027811000, Millipore Sigma).

## Mass spectrometry imaging
Initial lipid imaging was performed using a 4800 Plus MALDI TOF/TOF Analyzer equipped with a 200 Hz Nd:YAG pulse laser (355 nm) (Applied Biosystems SCIEX). The instrument was externally calibrated and operated in negative ion reflection mode to acquire data between mass ranges ($m/z$) of 500–1000 using DAN as a matrix. The number of laser shots per pixel was set at 50 and the raster distance between each pixel was set to 100 μm using the 4000 Series Explorer v5.5.3 and 4800 Imaging Tool v.3.2 (Applied Biosystem SCIEX, https://ms-imaging.org/wp/4000-series-imaging).

Additional lipid imaging was performed using a rapifleX MALDI Tissuetyper equipped with a Smartbeam 3D 10 kHz Nd:YAG (355 nm) laser (Bruker) and a timsTOF fleX MALDI-2 (TTF) equipped with a Smartbeam 3D 10 kHz Nd:YAG (255 nm) laser and microGRID (Bruker). The rapifleX MALDI Tissuetyper was operated in negative ion reflection mode to acquire data between mass ranges ($m/z$) of 500-2000 using 9AA or DAN as a matrix. Images were acquired with either 10 μm or 50 μm raster distance using 200 laser shots per pixels using flexControl 4.0 (Build 46) (Bruker). The timsTOF fleX was operated in negative ion transmission mode (in MALDI-1 mode) to acquire data between mass ranges ($m/z$) of 300-2500 using 9AA as matrix. Images were acquired with either 5 μm, 10 μm, or 20 μm raster distances, using 45, 150, and 200 laser shots per pixel, respectively with timsControl 4.1.8 (11f8cf17) (Bruker). microGRID was enabled for all TTF images. Laser power was adjusted on a tissue-to-tissue basis to optimize the signal.

For PC-MT imaging, the rapifleX MALDI Tissuetyper was operated in positive ion reflection mode to acquire data between mass ranges ($m/z$) of 1000-2000 using CHCA as a matrix. The number of laser shots per pixel was set at 200, and the raster distance between each pixel was set to 50 μm using flexControl 4.0 (Build 46) (Bruker).

Mass spectrometry imaging experiments were performed with up to three technical replicates from up to three biological replicates.

## Pre-lipid-GAMSI fluorescence imaging
~ 25 μm thick mouse brain slices were thawed, fixed with 4% PFA, and washed with 1x PBS three times for 5 min each time following the standard GAMSI sample preparation workflow described earlier. The fixed brain slices were incubated in a detergent-free blocking buffer [5% (v/v) normal goat serum (NGS) in 1x PBS] at room temperature for 6 hours. Next, the brain slices were incubated in the primary antibody (rabbit anti-calbindin, PA1-931, Thermo Fisher) solution (1:100 dilution with the detergent-free blocking buffer) overnight at room temperature. The brain slices were then washed with the blocking buffer four times for 30 min each time, and incubated in the secondary antibody (goat Alexa Fluor 568-conjugated anti-rabbit antibody, A11011, Thermo Fisher) solution (1:200 dilution with the detergent-free blocking buffer) overnight at room temperature. Finally, the brain slices were washed with 1x PBS (or the detergent-free blocking buffer) four times for 30 min each time and stored in 1x PBS. The brain slices were then gelled, homogenized, and expanded following the standard GAMSI sample preparation workflow described earlier. After mounting on an ITO-coated glass slide, pre-lipid-GAMSI fluorescence images were obtained using a Nikon spinning disk confocal system (CSU-W1, Yokogawa) with a 10× 0.45 NA air objective (Nikon) in wide-field mode. All fluorescence microscopy data were collected using NIS Elements AR v5.30.04 (Nikon). After pre-lipid-GAMSI fluorescence imaging, the mounted sample was coated with the matrix (DAN) as previously described and subject to mass spectrometry imaging using Bruker rapifleX.

## Post-lipid-GAMSI photocleavable mass-tag (PC-MT) modification
Post-lipid-GAMSI samples were washed with −80 °C acetone twice for 3 min each time to remove the matrix and then stained with the photocleavable mass-tags (PC-MTs) (AmberGen, Inc.) on-plate following the protocol provided by the vendor. Briefly, the samples were rehydrated in a washing sequence of 95% ethanol (EtOH) for 3 min, 70% EtOH for 3 min, and 50% EtOH for 3 min, followed by a 1x PBS wash for 10 min. Antigen retrieval was performed using 1x citrate buffer (pH = 6.0) [diluted from 10x citrate buffer (C9999-1000ML, Sigma Aldrich)] in a hot water bath for 1 hour at 95 °C, followed by cooling for 30 min at room temperature. The samples were then washed with 1x PBS for 10 min, and incubated with 5 mL of blocking buffer [2% (v/v) normal rabbit serum, 5% (w/v) bovine serum albumin in 1x PBS] for 1 hour at room temperature. Next, the samples were incubated with the PC-MT solution (2.5 μg/mL of each PC-MT in the blocking buffer) overnight at room temperature in a humidified chamber. PC-MTs myelin basic protein (MBP) (AP1001200, AmberGen, Inc.; $m/z$ = 1365.73) and Synapsin I (SYN-I) (AP1001204, AmberGen, Inc.; $m/z$ = 1482.77) were used. After PC-MT incubation, the samples were washed with 1x PBS three times for 5 min each time, then washed with 50 mM ammonium bicarbonate three times for 2 min each time, and dried in a Pyrex vacuum desiccator (Millipore Sigma) for 2 hours at room temperature. Finally, the PC-MTs were photocleaved in a UV (365 nm) chamber (AmberGen, Inc.) for 15 minutes before matrix application.

## Data processing
Data processing including region-of-interest determination, average mass spectrum extraction, and image generation, was conducted using MSiReader (ver. 1.02)[40] or SCiLS lab (ver. 2024a, Bruker). All images presented were normalized using Total Ions Count (TIC). Unless otherwise noted, all lipid assignments were made by comparing mass measurements to the LIPID MAPS database (www.lipidsmaps.com) with an allowed mass tolerance of $m/z$ = +/− 0.05. Lipid assignments from timsTOF fleX imaging were made using MetaboScape (Bruker) with an allowed mass tolerance of 2–5 ppm. Rigid registration across lipid and protein images was performed using TrackEM2 (ver. 1.0a 2012-07-04) plugin on ImageJ distribution Fiji (ver. 1.53t). Non-

rigid registration between fluorescence images and MS images was performed using BigWarp 9.0.0 plugin on ImageJ distribution Fiji (ver. 1.53t).

## Tandem mass spectrometry (MS/MS)

MS/MS was performed with the 4800 Plus MALDI TOF/TOF Analyzer (Applied Biosystems SCIEX) in negative ion reflection mode using DAN as a matrix. The instrument was operated in the 2 kV operation mode to allow unimolecular decay. Mass spectrometry data from 4800 Plus MALDI TOF/TOF Analyzer (Applied Biosystem SCIEX) were collected using 4000 Series Explorer v5.5.3 (Applied Biosystem SCIEX).

## Expansion isotropy analysis

To evaluate the spatial isotropy of sample homogenization and expansion, ~25 µm thick mouse brain slices were thawed, fixed with 3% PFA/0.1% GA, and washed with 1x PBS three times for 5 min each time following the standard GAMSI sample preparation workflow described earlier. The fixed brain slices were incubated in a detergent-free blocking buffer [5% (v/v) normal goat serum (NGS) in 1x PBS] at room temperature for 6 hours. Next, the brain slices were incubated in the primary antibody (rabbit anti-NF-200, N4142-.2 ML, Millipore Sigma) solution (1:100 dilution with the detergent-free blocking buffer) overnight at room temperature. The brain slices were then washed with the blocking buffer four times for 30 min each time, and incubated in the secondary antibody (goat Alexa Fluor 568-conjugated anti-rabbit antibody, A11011, Thermo Fisher) solution (1:200 dilution with the detergent-free blocking buffer) for 2 days at room temperature. Finally, the brain slices were washed with 1x PBS (or the detergent-free blocking buffer) four times for 30 min each time and stored in 1x PBS.

Pre-expansion fluorescence images were obtained using a Nikon spinning disk confocal system (CSU-W1, Yokogawa) with a 40×1.15 NA water immersion objective (Nikon). After pre-expansion imaging, the brain slices were gelled, homogenized, and expanded following the standard GAMSI sample preparation workflow described earlier. The expanded brain slices were then imaged using the same confocal microscope system and objective across regions of interest (ROIs) corresponding to those obtained from the pre-expansion imaging. All fluorescence microscopy data were collected using NIS Elements AR v5.30.04 (Nikon). Lastly, registration of pre- and post-expansion images, generation of the vector deformation field, and calculation of the root mean square (r.m.s) error for all the point-to-point measurements were performed using a custom MATLAB code as previously described[41].

## Lipid retention

For (non-expanded) fresh-frozen samples, ~25 µm thick mouse brain slices were thawed to room temperature and immediately stained with HCS LipidTOX™ Red Phospholipidosis Detection Reagent (H34351, Thermo Fisher) (1:200 dilution with 1x PBS) overnight at room temperature for phospholipid labeling. The brain slices were then fixed with 3% PFA/0.1% GA at room temperature for 2 hours to maintain the tissue integrity. Finally, the fixatives were removed, and the brain slices were washed with 1x PBS three times for 20 min each time.

For expanded samples, ~25 µm thick mouse brain slices were thawed, fixed, gelled, and homogenized following the standard GAMSI sample preparation workflow. The samples were then incubated with the same HCS LipidTOX™ Red Phospholipidosis Detection Reagent (1:200 dilution with 1x PBS) overnight at room temperature and DAPI solution (100 ng/mL in 1x PBS) for 2 hours. Finally, the samples were expanded with purified water three times for 20 min each time.

Wide-field epifluorescence images of both the fresh-frozen and the expanded lipid-stained brain slices were obtained with a 10× 0.45 NA air objective (Nikon) with 2 ms exposure. All fluorescence

microscopy data were collected using NIS Elements AR v5.30.04 (Nikon). Total pixel values per pre-expansion unit tissue area of the white matter, molecular layer, and granular layer of the cerebellum were measured to represent the fluorescence intensity per unit tissue area.

## Comparison of trypsin and proteinase K (proK) digestion and evaluation of the surfactant effect

~40 µm thick mouse brain slices (from one animal, perfused and fixed with 4% PFA and sectioned using a Leica VT1200 vibratome) were used to evaluate different digestion conditions for the sample homogenization step. For trypsin digestion (without surfactants), the brain slice was subject to the standard GAMSI sample preparation with trypsin digestion. Briefly, the fixed brain slice was incubated in the detergent-free blocking buffer [5% (v/v) normal goat serum (NGS) in 1x PBS] at room temperature for 6 hours. Next, the brain slice was incubated in the primary antibody (rabbit anti-NF-200, N4142-.2 ML, Millipore Sigma) solution (1:100 dilution with the detergent-free blocking buffer) overnight at room temperature, and then washed with the blocking buffer four times for 30 min each time. The brain slice was then incubated in the secondary antibody (goat Alexa Fluor 568-conjugated anti-rabbit antibody, A11011, Thermo Fisher) solution (1:200 dilution with the detergent-free blocking buffer) for 2 days at room temperature, washed with 1x PBS or the blocking buffer four times for 30 min each time, and stored in 1x PBS. Finally, the brain slice was gelled, homogenized with the trypsin digestion buffer, expanded, and imaged following the standard GAMSI sample preparation workflow. Pre- and post-expansion fluorescence images were obtained using a Nikon spinning disk confocal system (CSU-W1, Yokogawa) with a 4 × 0.20 NA air objective (Nikon). All fluorescence microscopy data were collected using NIS Elements AR v5.30.04 (Nikon).

For proK digestion (with surfactants), brain slice was subject to the proExM protocol[29,30]. Briefly, the fixed brain slice was permeabilized with 0.1% (w/v) Triton X-100 in 1x PBS for 15 min and incubated in a blocking buffer [5% (v/v) normal goat serum (NGS) and 0.1% (w/v) Triton X-100 in 1x PBS] at room temperature for 6 hours. Next, the brain slice was incubated in the primary antibody (rabbit anti-NF-200, N4142-.2 ML, Millipore Sigma) solution (1:100 dilution with the blocking buffer) overnight at room temperature and washed with the blocking buffer four times for 30 min each time. The brain slice was then incubated in the secondary antibody (goat Alexa Fluor 568-conjugated anti-rabbit antibody, A11011, Thermo Fisher) solution (1:200 dilution with the blocking buffer) overnight at room temperature, washed with 1x PBS or the blocking buffer four times for 30 min each time, and stored in 1x PBS. Finally, the brain slice was gelled, homogenized with the proK digestion buffer [proK (8 units/mL) in 50 mM Tris (pH 8), 1 mM EDTA, 0.5% Triton X-100, 1 M NaCl], expanded, and imaged following the proExM protocol. Pre- and post-expansion images were obtained using the previously described Nikon spinning disk confocal system with a 4× 0.20 NA air objective in wide-field mode. All fluorescence microscopy data were collected using NIS Elements AR v5.30.04 (Nikon).

To evaluate the surfactant effect, ~25 µm mouse brain slices were subjected to standard GAMSI sample preparation. The samples were then digested separately for 2 days at 37 °C in four different homogenization conditions: (1) proK with surfactants [proK (8 units/mL) in 50 mM Tris (pH 8), 1 mM EDTA, 0.5% Triton X-100, 1 M NaCl], (2) trypsin with surfactants [0.25% (w/v) trypsin in 50 mM Tris (pH 8), 1 mM EDTA, 0.5% Triton X-100, 1 M NaCl], (3) proK without surfactants [proK (8 units/mL in 1x PBS)] and (4) trypsin without surfactants [0.25% (w/v) trypsin in 1x PBS]. After expansion, wide-field images of the samples were obtained using the previously described Nikon spinning disk confocal system with a 4× 0.20 NA air objective, 300 ms exposure, and oblique white-light LED illumination. All optical microscopy data were collected using NIS Elements AR v5.30.04 (Nikon).

## Statistics and reproducibility

No statistical method was used to predetermine sample size. No data were excluded from the analyses. The experiments were not randomized. The investigators were not blinded to allocation during experiments and outcome assessment. For reproducibility, all experiments were repeated independently at least three times, unless otherwise noted.

## Reporting summary

Further information on research design is available in the Nature Portfolio Reporting Summary linked to this article.

## Data availability

The mass spectrometry data generated in this study have been deposited in the public MassIVE repository under accession code MSV000094777. The raw data for Fig. 5b (over 70 GB) is too large to be shared over MassIVE and is available upon request. Requests for data will be fulfilled within two weeks. Source data are provided in this paper.

## Code availability

Custom scripts used for data analysis are available on GitHub at https://github.com/HoraceChan99/NonRigidReg.git.

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

## Acknowledgements

We thank F. Tobias, S. Shafaie, and the Northwestern University IMSERC facility for assistance with mass spectrometry imaging. We thank the Chicago Biomedical Consortium (CBC) for access to core mass spectrometry facilities. We thank W. Wang for assistance with hydrogel reagent preparation, C.-C. (J.) Yu for assistance with expansion isotropy analysis, and S. Kwon for assistance with scientific visualization and illustration. R.G. acknowledges funding support from US NIH DP2MH136390, US NIH UG3MH126864, Searle Scholars Program, McKnight Technological Innovations in Neuroscience Award, and the University of Illinois Chicago Startup Fund. S.M.C. acknowledges funding support from US NIH R01NS114413, US NIH R01NS124784, and US NSF CAREER Award 2143920.

## Author contributions

Conceptualization: R.G. and S.M.C. Methodology: Y.H.C. and R.G. Investigation: Y.H.C., K.C.P., D.P.-J., M.C.H., N.T., J.L.F., E.Y., and R.G. Formal analysis: Y.H.C. Software: Y.H.C. Visualization: Y.H.C. and R.G. Funding acquisition: R.G. and S.M.C. Project administration: R.G. Supervision: R.G. and S.M.C. Writing – original draft: R.G. and Y.H.C. Writing – review & editing: Y.H.C., K.C.P., D.P.-J., N.T., J.L.F., E.Y., S.M.C., and R.G.

## Competing interests

R.G. is a co-inventor of multiple patents related to expansion microscopy. The other authors declare that they have no competing interests.
