## [Peer Review File · Nature Communications]

Reviewers' Comments:

Reviewer #1:

Remarks to the Author:

The authors present a powerful methodology for increasing effective spatial resolution in mass spectrometry imaging (MSI) through gel-assisted sample expansion, in the context of matrix assisted laser desorption ionization (MALDI) MSI. Typical MALDI workflows are limited to 5-50 micron spatial resolution, but with this approach sub-micron effective resolution can be achieved without special mass spectrometry hardware. Thus, this methodology has significant potential to enhance capabilities of MSI in the real world.

The authors perform a rigorous assessment of the uniformity (isotropy) of sample expansion, as well as a quantitative assessment of (non-specific) phospholipid retention through their sample preparation steps. This demonstrates that the approach is yielding expanded samples which are still spatially and, at least with regards lipids in general, chemically representative of the original samples.

Additionally, experiments are performed across several different instruments, with different matrices, demonstrating the reproducibility and potential transferability of this method. A brief example of protein-imaging is also presented, though this example is much less developed. The MS instruments used are all of medium mass resolution (TOF-type instruments), which may be unable to detect and resolve much of the true underlying lipid complexity. Future work should explore this further with higher resolution mass analyzers (e.g. Orbitrap, ICR).

Overall, I think this methodology has significant potential for the community and will be well received by the MSI community, as well as serving as a useful example for the wider biochemical community of how molecular-level MSI can be useful at cellular and sub-cellular resolution.

I have several specific points which I encourage the authors to address:

1) On line 91 and throughout, the authors refer to 'sample homogenization'. To me, this means homogenization of the tissue in such a way as to lose all spatial information (e.g. bead-beating or ball milling) – I'm sure the authors don't mean that, though. Can they confirm this is the correct term, and provide a more specific definition given the more general nature of this journal?

2) Line 120-122/Figure 2b – '...replicated nearly all the major lipid peaks'. This statement reads quite subjective/non-quantitative, and a visual inspection of the Figure 2b (as well as Sup FigS1a-c) shows that there are quite significant differences in the MS profiles between fresh-frozen and GAMSII prepped samples. Not only are there features present in one spectrum which aren't present in the other, the relative signal intensities are significantly different – which does not seem to be commented on here.

a. I would like to see a more quantitative analysis – e.g. how many signals are in common vs different, how many annotations are in common vs different, and how does the signal intensity change for each annotation?

b. I note Figure 2b has y-axis top scale at 0-22,000 and bottom scale 0-6,000 counts. (~3.67x), which I assume is related to the ~3x tissue expansion?

3) Throughout, I think the topic of sensitivity is under-discussed. One of the major limitations in small-pixel MALDI is sensitivity – subcellular MALDI has best been demonstrated using transmission-mode geometry, but at such small sizes (<1um) post-ionization ('MALDI-2') is important to boost sensitivity sufficiently. Here, by stretching the tissue 3x-6x, presumably dilution of analytes becomes a sensitivity issue in the same regard? Or, does it not because the total concentration of analytes is not the issue in MALDI but just the poor desorption/ionization efficiency of the MALDI process? Either way – I would like to see this point highlighted more clearly as a potential limitation of this approach.

a. If 'analyte dilution' through expansion is a real issue, is this remedied through using thicker sections of tissue? E.g. does a 4x expansion on a 10um-thick section yield less sensitivity than a 4x expansion on a 20um-thick section?

4) The authors should confirm if MALDI-2 was enabled when using the timsTOF Flex instrument, and if any comparison between MALDI-1 and MALDI-2 was made on this instrument to benchmark the sensitivity gains afforded by MALDI-2 and how this may offset sensitivity losses through tissue expansion?

5) Line 113/114 – The authors report 84% and 82% preserved lipids, however the assay is non-specific. It would be interesting to know if their MS results can provide information about the preservation of lipids. Specifically, is the loss non-specific, or are specific types of lipid being persevered more efficiently than others?

a. I note that the authors highlight that MSI is 'quantitative' several times throughout the text, which is a potentially controversial statement in itself, so if they chose not to use the MSI quantitatively here, this should be mentioned and rationalized.

6) Throughout, the authors switch between referring to the true instrument pixel size (i.e. the combination of laser focus and stage steps) and 'effective' pixel size (which I presume to be the calculated size of a pixel based on the hypothetical unexpanded tissue? I think some clarification of meaning here would be helpful to the readers.

7) Figure 3 – The tissue orientations in panels A and B are not presented the same way (it seems they're rotated 90° and flipped vertically). It may be preferable to present them in the same orientation.

8) Figure 4a – I would find it helpful to see a little more clarification that the ion images presented are for the photo-cleaved anti-body mass tag, and not for the protein itself?

Reviewer #2:

Remarks to the Author:

Review comments

Mass Spectrometry Imaging (MSI) is an important tool for examining native biomolecules in specimens, but its application at the single-cell and subcellular levels has been restricted due to its spatial resolution limits. To address this, the authors developed a new method called Gel-Assisted Mass Spectrometry Imaging (GAMSI). The difference between GAMSI and standard MSI is that the GAMSI employs expansion microscopy in sample preparation. Sample expansion enhanced the spatial resolution of lipid MALDI-MSI by about 3-6 times, reaching the sub-micrometer scale. This enhancement is achieved without altering the existing mass spectrometry hardware or analysis process. The authors successfully demonstrated the resolution improvement by GAMSI compared to standard MSI lipidomics of mouse brain tissue sections. They further extended the method to multiplexed MS imaging of lipid and several proteins targeted by antibody conjugated with photocleavable TM probes that can be detected by MS after photocleavage. Major revision is recommended because the chemical foundation of the principle of GAMSI is unclear.

Specific comments

1. The chemical principle behind GAMSI lipidomics is not well explained. In order to isotropically expand the structure of tissues made of lipids, the lipids must be anchored to the hydrogel. However, the method used only Acryloyl-X (acrylic NHS ester) as the anchor, which reacts only with proteins. This raises an important question: How are lipids anchored to the hydrogel? The authors must address this critical issue to fully explain the principle of GAMSI lipidomics.

2. The workflow presented in Figure 1 is somewhat misleading as it fails to include the crucial step of antibody labeling for targeted proteome imaging, even though the main text mentions that photocleavage occurs on the antibodies conjugated with Photocleavable mass-tag (PC-MT) modification. Due to the lack of this step, it is unclear which step is responsible for lipid releasing and which one for antibody releasing. Therefore, Figure 1 should be updated to include the antibody labeling step and the magnified insets should display the antibodies to avoid any confusion.

3. The content in lines 62-68 of the results section needs to be relocated to the introduction section for better clarity. Ref 19 is the most relevant reference to this work because it also combines MSI and expansion microscopy for high resolution. It should be described in detail in the introduction.

4. The authors should provide a more detailed introduction on the working principle of expansion

microscopy and its various applications in proteomics, lipidomics, and multiomics.

5. In line 74, it is unclear how "the target biomolecules can be retained via physical interaction with the hydrogel itself". Further explanation or an example is needed.

6. In line 77, explain how "the target biomolecules can be chemically tethered to the hydrogel polymer chains using a cleavable small molecule linker".

7. The dynamic range of the lipidome detection should be discussed. How many different types of lipids are detected? How different is the GAMS1 lipidomics from the standard non-expansion MSI?

8. In the method section, provide the catalog numbers of the PC-MTs. If they are customized orders, provide sequences of the MTs. Provide the chemical name of the PC linker, eg. nitrobenzyl. Briefly describe the antibody conjugation method. Catalog numbers of all other chemical and biochemical reagents should be included.

Reviewer #3:

Remarks to the Author:

This paper describes an imaging workflow to interrogate molecular species using mass spectroscopy on samples that have been expanded in size by coupling the tissue to an absorbant hydrogel that swells in size. The goal is to get sub-micrometer resolution to overcome the spatial limitations of point scanning in mass spectroscopy. Once this approach reaches subcellular resolution it may be quite useful in multiplexing many molecular labels on the same tissue sample, a potentially valuable approach to add to the armamentarium that presently relies heavily on the very powerful approaches of spatial transcriptomics.

The main problem with this paper is that the promise of sub-micrometer resolved images is not fully realized. None of the images show cellular, much less subcellular resolution. Fig 5b shows phospholipids as white, but a blow-up of that image shows phosphatidic acid (according to the label in the figure) as black- but the images look the same, so something is amiss. In either case however, there are no clear cellular details. The granule cell layer which is densely packed with neurons and neuropil so there is little to see that would require submicron resolution. Perhaps the Purkinje cell layer which has substantially larger cells and cells that are in a single line would be a better target for this proof of concept. The arrow heads may be pointing at some cell type as suggested by the authors, but the fluorescence image provided as evidence is grossly subpar. The two channels should be shown separately. I would have expected to see some single myelinated axons in the granule cell layer- given the submicron resolution but none were apparent.

Lastly the power of this approach requires showing a high degree of multiplexing. Two labels is not sufficient and showing more would be more compelling.

In sum the technique needs to resolve cellular or subcellular details to be of value in brain samples and multiple protein labels (>2) at the same time would also be helpful.

We thank the Editor and the Reviewers for the detailed suggestions on how we could improve our manuscript. We have performed additional experiments and analyses to better understand the chemical basis of GAMSIs, to demonstrate the (sub)cellular imaging capability of GAMSIs by focusing on a distinct cell layer as suggested by the Reviewer, and to showcase the multiplexing power of GAMSIs with three protein targets. We have revised the main text/figures as well as the supplementary materials (with major changes highlighted in color) to address all the questions raised by the Reviewers. We hope the updated version of our manuscript is acceptable at Nature Communications.

Reviewer #1:

The authors present a powerful methodology for increasing effective spatial resolution in mass spectrometry imaging (MSI) through gel-assisted sample expansion, in the context of matrix assisted laser desorption ionization (MALDI) MSI. Typical MALDI workflows are limited to 5-50 micron spatial resolution, but with this approach sub-micron effective resolution can be achieved without special mass spectrometry hardware. Thus, this methodology has significant potential to enhance capabilities of MSI in the real world.

The authors perform a rigorous assessment of the uniformity (isotropy) of sample expansion, as well as a quantitative assessment of (non-specific) phospholipid retention through their sample preparation steps. This demonstrates that the approach is yielding expanded samples which are still spatially and, at least with regards lipids in general, chemically representative of the original samples.

Additionally, experiments are performed across several different instruments, with different matrices, demonstrating the reproducibility and potential transferability of this method. A brief example of protein-imaging is also presented, though this example is much less developed. The MS instruments used are all of medium mass resolution (TOF-type instruments), which may be unable to detect and resolve much of the true underlying lipid complexity. Future work should explore this further with higher resolution mass analyzers (e.g. Orbitrap, ICR).

Overall, I think this methodology has significant potential for the community and will be well received by the MSI community, as well as serving as a useful example for the wider biochemical community of how molecular-level MSI can be useful at cellular and sub-cellular resolution.

We are grateful to the Reviewer for highlighting the significance of our work for the mass spectrometry community. The Reviewer has also noted the current limitation of MSI, as well as the innovative aspect of our approach to overcome this limitation. We agree with the Reviewer on the suggested future directions, which include combining GAMSIs with high mass-resolution MS instrument. We seek to pursue such future projects by sharing our method and collaborating with the mass spectrometry community.

I have several specific points which I encourage the authors to address:

1) On line 91 and throughout, the authors refer to 'sample homogenization'. To me, this means homogenization of the tissue in such a way as to lose all spatial information (e.g. bead-beating or ball milling) – I'm sure the authors don't mean that, though. Can they confirm this is the correct term, and provide a more specific definition given the more general nature of this journal?

We thank the Reviewer's comments on the usage of "sample homogenization". We recognize the term has a distinctive meaning (= disruption or disintegration of tissue via, for example, bead-beating and ball-milling) when used in the context of MS sample preparation, and we confirm this is NOT what we meant. As the Reviewer pointed out, when "sample homogenization" is used in the context of expansion microscopy, it refers to a (bio)chemical treatment step that reduces (= softens) the structural integrity of the gelled specimen while (largely) preserving its original spatial and molecular information. Following the Reviewer's suggestion, we have included a clearer definition of "sample homogenization" (within the context of expansion microscopy) in the revised main text (Page 4, Line 123-127).

2) Line 120-122/Figure 2b – '...replicated nearly all the major lipid peaks'. This statement reads quite subjective/non-quantitative, and a visual inspection of the Figure 2b (as well as Sup FigS1a-c) shows that there are quite significant differences in the MS profiles between fresh-frozen and GAMS1 prepped samples. Not only are there features present in one spectrum which aren't present in the other, the relative signal intensities are significantly different – which does not seem to be commented on here.

We thank the Reviewer for pointing this out. Based on the GAMS1 and fresh-frozen MS spectra, we have performed an in-depth comparative analyses to assess the similarity as well as difference in their MS profiles (see below). We included the results of our analyses and associated discussions in the revised main text (Page 4, Line 154-168; Page 5, Line 189-199).

a. I would like to see a more quantitative analysis – e.g. how many signals are in common vs different, how many annotations are in common vs different, and how does the signal intensity change for each annotation?

To address the Reviewer's comment, we have analyzed the common vs. different lipid peaks from the averaged mass spectra (obtained using AB SCIEX 4800 MALDI-TOF) of the fresh-frozen and GAMS1-processed mouse cerebellum sample. First, we removed peaks with <3:1 signal-to-noise ratio (SNR) and labeled the rest of the peaks (with $m/z = 650-925$) as "detected peaks". Next, we used the LIPID MAPS database to automatically assign the detected peaks (with +/- 0.05 m/z tolerance), and created a list of annotated lipids for each sample. Finally, we compared this list and tabulated lipids that are common vs. different across the two samples. While we do not have an MS/MS validation for all the annotated peaks, we have postulated that such an analysis can still provide a more quantitative view of the extent of lipid coverage in the two samples. In addition, we note that a more systematic quantitative MS analysis using different tissue types and technical/ biological replicates is necessary as future studies to fully characterize the MS profiles of the two samples (see later discussion).

The results of our lipid profile analysis are shown in a Venn diagram in **Fig. S6**. In summary, 58 lipids were detected as common peaks between the fresh-frozen and GAMSJ sample. 60 and 31 lipids were detected only in the fresh-frozen and the GAMSJ sample, respectively. The 58 common lipids corresponded to ~49% of all the lipids (= 58 + 60) detected for the fresh-frozen sample.

During this analysis, we noticed that a majority of the fresh-frozen-only peaks (= lipids detected in the fresh-frozen sample but not in the GAMSJ sample) were of low relative intensities within its MS spectrum. As we focus on the relative quantification aspect of MSI in this manuscript, we went ahead and analyzed the difference in the relative signal intensities of the two spectra to evaluate this effect (and to address the Reviewer's question on the signal intensity). In short, we normalized the MS spectra of both samples using their respective $m/z = 885.54$ peak (= an abundant phosphoinositide (PI) peak found in both samples) to obtain the relative signal intensity for all the detected peaks. We then calculated the changes in the relative signal intensity for each detected peak by subtracting one spectrum from the other. As result, we found that a majority (>80%) of the common peaks had <20% changes in its relative signal intensity. We also found that a majority (>95%) of fresh-frozen-only peaks (= lipids detected in the fresh-frozen sample but are missed in the GAMSJ sample) were of <20% relative signal intensity, which confirmed our initial observation. Combined, these results suggest that some of the fresh-frozen-only lipids could have become undetectable in the GAMSJ sample run because of the low relative signal intensity. We speculated this could have been caused by low instrument sensitivity and dilution of the analytes in the GAMSJ sample (see our discussion in 2b) on the "analyte dilution effect").

Given our hypothesis, we postulated that loss of the fresh-frozen-only peaks in the GAMSJ sample could be (partially) remedied by switching to a MALDI-TOF mass spectrometer with higher sensitivity. Indeed, when we performed the same MS profile analysis with MS spectra obtained using Bruker rapifleX, we found more common lipids existed between the fresh-frozen and GAMSJ sample (**Fig. S12**). Furthermore, the overall number of lipids detected for both samples increased drastically. In fact, more GAMSJ-only peaks (145) were detected than the fresh-frozen-only peaks (90). This is not surprising given that fresh-frozen and GAMSJ sample went through different sample preparation processes and that the GAMSJ process may have uncovered lipids that are hard to access by conventional fresh-frozen sample preparation.

Finally, in response to the Reviewer's subsequent question on GAMSJ's chemical perturbation, we have created a pie diagram to illustrate the chemical breakdown (i.e., lipid types) of the fresh-frozen vs. GAMSJ lipid peaks found in the AB SCIEX 4800 spectra (**Fig. S6b**). As result, we found that the chemical composition of detected lipids stayed largely consistent across the fresh-frozen vs. common vs. GAMSJ lipid peaks. One quick validation we did was to calculate the percentage of phospholipids possessing a primary amine head group (= PE + PS) within each category. To our surprise, this value also remained consistent across the fresh-frozen (~48%) vs. common (~48%) vs. GAMSJ (~49%) categories. Provided that the GAMSJ-processed sample goes through additional chemical processes that crosslink primary amines (such as the

light chemical fixation step and the NHS-ester-based membrane protein anchoring step), we initially speculated a significant suppression of these lipid species in the GAMSJ sample. However, this analysis result indicates that a large portion of PE and PS remain chemically unperturbed after these steps in the GAMSJ sample preparation.

To summarize our findings, we have included associated analyses and discussions in the new supplementary figures (**Fig. S6** and **Fig. S12**) and the revised main text (Page 4, Line 154-168; Page 5, Line 189-199).

Lastly, we note that to fully characterize the lipid coverage and quantify its perturbation under GAMSJ, additional MS studies are necessary. In particular, the effect of the chemical fixation and NHS-ester-based anchoring steps will need to be evaluated (separately) with additional technical and biological replicates. In addition, such characterizations need to be performed on other tissue types (e.g., from other organs or other modal organisms) to increase the lipid type and diversity. We hope to tackle such analysis and quantification in the future through continuing collaboration with the MS community.

b. I note Figure 2b has y-axis top scale at 0-22,000 and bottom scale 0-6,000 counts. (~3.67x), which I assume is related to the ~3x tissue expansion?

We agree with the Reviewer that the observed decrease in the signal counts could be attributed to (at least partially, see below) the expansion-induced dilution of the analytes. In 3x expanded GAMSJ samples, for example, the number of analytes per unit sample area are decreased by 9x (= proportional to the square of the expansion factor). This decrease of analytes per unit area may have contributed to the observed decrease in the absolute signal.

In addition to this “analyte dilution effect”, however, we note that there might be other factors affecting the final signal counts (and possibly compensating for the analyte dilution effect) of the GAMSJ sample. For example, due to the expansion, non-analytes are also diluted. This makes the analyte signals more prominent by suppressing the “matrix effect”. Additional effects like this may explain why the observed decrease in the signal counts was not as drastic as 9x.

In summary, we agree that the observed decrease in the signal counts can be attributed to sample expansion, but there could be additional factors that have compensated for this effect.

3) Throughout, I think the topic of sensitivity is under-discussed. One of the major limitations in small-pixel MALDI is sensitivity – subcellular MALDI has best been demonstrated using transmission-mode geometry, but at such small sizes (<1um) post-ionization (‘MALDI-2’) is important to boost sensitivity sufficiently. Here, by stretching the tissue 3x-6x, presumably dilution of analytes becomes a sensitivity issue in the same regard? Or, does it not because the total concentration of analytes is not the issue in MALDI but just the poor desorption/ionization efficiency of the MALDI process? Either way – I would like to see this point highlighted more clearly as a potential limitation of this approach.

We thank the Reviewer for the detailed comments and suggestions. In response, we have addressed pending questions (see below) and included a detailed discussion on the sensitivity and limitation of our approach in revised main text (Page 7, Line 258-267). We have also included potential remedies to this limitation (such as MALDI-2 and changing the sample geometry) as suggested by the Reviewer.

a. If 'analyte dilution' through expansion is a real issue, is this remedied through using thicker sections of tissue? E.g. does a 4x expansion on a 10um-thick section yield less sensitivity than a 4x expansion on a 20um-thick section?

In principle, we agree with the Reviewer that thicker tissue sections can provide higher signal counts (and thus better imaging sensitivity) for GMSI. In the GMSI sample preparation, the gelled sample is expanded and then collapsed to a planar geometry. During the subsequent matrix application and recrystallization steps, the number of analytes per unit sample area should increase in proportion to the tissue section thickness, provided that analytes can be extracted by the matrix across the entire thickness of the collapsed GMSI sample. However, the sample conductivity may also decrease (and the sensitivity will decrease) if the tissue section is too thick. Therefore, there should be a limit to how thick we can increase the tissue sections without affecting the sensitivity negatively.

Thus far, however, we have not obtained consistent experimental results to back up or reject this hypothesis. This is because we haven't been able to prepare the sample exactly the same way using our setup (e.g., applying matrix at a consistent thickness/amount). In addition, the instrumental sensitivities of the MALDI-TOF mass spectrometers have not been consistent across our experiments either (as they are owned by a core facility or are close to the end of its operation). Combined, these factors have made absolute comparisons of imaging sensitivities challenging at the moment. We look forward to solving these issues through additional optimizations and collaborations.

To clarify these points, we have included a detailed discussion in the revised main text (Page 7, Line 263-267).

4) The authors should confirm if MALDI-2 was enabled when using the timsTOF Flex instrument, and if any comparison between MALDI-1 and MALDI-2 was made on this instrument to benchmark the sensitivity gains afforded by MALDI-2 and how this may offset sensitivity losses through tissue expansion?

We used the standard MALDI-1 imaging mode for all the timsTOF fleX imaging shown in the manuscript. We have added this clarification to the figure caption of **Fig. 5** (Page 22, Line 718-720).

As the Reviewer kindly pointed out, post-ionization (e.g., MALDI-2 imaging mode on timsTOF fleX) is a promising approach to remedy the analyte dilution effect. At the moment, we haven't explored MALDI-2 imaging extensively with GMSI samples because a successful run will

require additional optimizations of the matrix and imaging conditions (and in addition, MALDI-2 setups have limited accessibility to most of the facilities and labs). However, this is a direction we would like to pursue and subsequently share our findings with the community on.

To clarify these points, we have included a similar discussion in the revised main text (Page 7, Line 262-263).

5) Line 113/114 – The authors report 84% and 82% preserved lipids, however the assay is non-specific. It would be interesting to know if their MS results can provide information about the preservation of lipids. Specifically, is the loss non-specific, or are specific types of lipid being persevered more efficiently than others?

We thank the Reviewer for this important question. As discussed earlier, we analyzed the chemical breakdown (= lipid types) of the detected lipid peaks in the fresh-frozen and GAMSII sample (**Fig. S6b**). As result, we found that lipid composition stayed largely constant across the fresh-frozen vs. common vs. GAMSII peaks. In particular, the ratio of phospholipids possessing a primary amine head group (= PE + PS) remained consistent across the fresh-frozen (~48%) vs. common (~48%) vs. GAMSII (~49%) categories. As described earlier, the GAMSII sample goes through more chemical processes, such as an additional chemical fixation step (e.g., PFA fixation) and an NHS-based membrane protein anchoring step, compared to the fresh-frozen sample. In fact, both of these steps serve to covalently anchor PE and PS to other molecules or the hydrogel network itself, and hence can hinder their ionization in the subsequent imaging steps. The fact that the PE and PS peaks were not specifically suppressed in the GAMSII sample suggests that a majority of these phospholipids are not as chemically affected as we speculated. These analyses provide preliminary yet critical insights into the chemical perturbation of the GAMSII sample preparation. As mentioned, we have included the related discussions in the revised main text (Page 4, Line 154-168),

a. I note that the authors highlight that MSI is ‘quantitative’ several times throughout the text, which is a potentially controversial statement in itself, so if they chose not to use the MSI quantitatively here, this should be mentioned and rationalized.

Following the Reviewer’s suggestion, we have removed related descriptions about the “(absolute) quantitative” capability of MSI to avoid potential controversy. Instead, we have included a statement that this work is based on the relative quantification aspect of MSI because it is more widely used by the field and does not require additional standards or calibrations (Page 3, Line 75-77).

6) Throughout, the authors switch between referring to the true instrument pixel size (i.e. the combination of laser focus and stage steps) and ‘effective’ pixel size (which I presume to be the calculated size of a pixel based on the hypothetical unexpanded tissue? I think some clarification of meaning here would be helpful to the readers.

The Reviewer is right about our definition of the two terms. The instrument pixel size refers to the physical pixel size determined by the instrument specs such as the laser spot size, raster distance, and the sample step size. The effective pixel size refers to the calibrated pixel size by dividing the instrument pixel size by the sample expansion factor. For example, when a 4x expanded sample is imaged with 50 μm instrument pixel size, the effective pixel size (which pertains to the original biological scales, hence we created this term) is 12.5 μm ($= 50 \mu\text{m}/4$). Following the Reviewer's suggestion, we have provided a clear definition of these two terms in the figure caption of **Fig. 2** (where one of the terms first appears in the main text) (Page 18, Line 684-686).

7) Figure 3 – The tissue orientations in panels A and B are not presented the same way (it seems they're rotated 90° and flipped vertically). It may be preferable to present them in the same orientation.

We thank the Reviewer for the suggestion. We have revised **Fig. 3** so that the tissue sections are now presented in the same orientation.

8) Figure 4a – I would find it helpful to see a little more clarification that the ion images presented are for the photo-cleaved anti-body mass tag, and not for the protein itself?

We thank the Reviewer for the suggestion. We have included clarifications on the nature of protein ion images (= mass reporters of the photocleaved anti-body mass tags) in the figure caption of **Fig. 4** (Page 20, Line 697-700).

Reviewer #2:

Mass Spectrometry Imaging (MSI) is an important tool for examining native biomolecules in specimens, but its application at the single-cell and subcellular levels has been restricted due to its spatial resolution limits. To address this, the authors developed a new method called Gel-Assisted Mass Spectrometry Imaging (GAMSI). The difference between GAMSI and standard MSI is that the GAMSI employs expansion microscopy in sample preparation. Sample expansion enhanced the spatial resolution of lipid MALDI-MSI by about 3-6 times, reaching the sub-micrometer scale. This enhancement is achieved without altering the existing mass spectrometry hardware or analysis process. The authors successfully demonstrated the resolution improvement by GAMSI compared to standard MSI lipidomics of mouse brain tissue sections. They further extended the method to multiplexed MS imaging of lipid and several proteins targeted by antibody conjugated with photocleavable TM probes that can be detected by MS after photocleavage. Major revision is recommended because the chemical foundation of the principle of GAMSI is unclear.

We thank the Reviewer for commenting on the significance of our work for single-cell/subcellular multi-omics. The Reviewer has also recognized the technical advances of our work, which

includes 3-6 fold increase of MALDI-MSI spatial resolution and application to multiplexed lipid and protein imaging. To address the Reviewer's comments, we have included detailed analyses and discussions on the chemical principle of GAMSII in the revised manuscript.

Specific comments

1. The chemical principle behind GAMSII lipidomics is not well explained. In order to isotropically expand the structure of tissues made of lipids, the lipids must be anchored to the hydrogel. However, the method used only Acryloyl-X (acrylic NHS ester) as the anchor, which reacts only with proteins. This raises an important question: How are lipids anchored to the hydrogel? The authors must address this critical issue to fully explain the principle of GAMSII lipidomics.

We thank the Reviewer for the comment. In fact, the same question has concerned us for some time. To better understand the chemical principle behind GAMSII, we have been looking into literature as well as experimental evidence of our own.

In short, our current hypothesis is that most of the endogenous lipids detected by GAMSII are tethered to the swellable hydrogel network via a non-covalent interaction with the anchored proteins. In fact, membrane proteins constitute as much as ~25% of all proteins and are a major component of biological membranes. The non-covalent interactions between membrane proteins and lipids are known to play an important role in stabilizing biological membranes by keeping the membrane proteins in place but also allowing them to move around/across the membrane when necessary. There have been extensive experimental and computational studies on the chemical and physical nature of these interactions over the past few decades (see, for example, Jiang et al., *Nat. Commun.* **13**, 7373 (2022) and Muller et al., *Chem. Rev.* **119**, 6086-6161 (2019)).

Of all the non-covalent interactions, hydrophobic interactions ($\sim k_B T$) are known to be a major binding force between the membrane proteins and lipids. Previous studies have shown that such interactions can extend as far as ~10 nm from the membrane proteins (see Jiang et al., *Nat. Commun.* **13**, 7373 (2022)). This suggests that membrane lipids could be tethered around each protein (to up to ~10 nm in diameter) when the membranes are dissociated. In the GAMSII sample preparation, the membrane proteins are covalently anchored to the hydrogel polymer network via the AcX linker. Hence, a portion (if not all) of the membrane lipid constituents can be preserved under this mechanism, provided that no disruption of this hydrophobic interactions occurs during the sample preparation.

In conventional expansion microscopy protocols (e.g., the original ExM and the proExM protocol), however, high concentration surfactants are typically used in the sample homogenization/digestion step. In this case, surfactants, which are amphiphilic molecules, serve to disrupt the intra-molecular hydrophobic interaction of membrane lipids in the aqueous environment. To validate our hypothesis on the non-covalent nature of lipid retention, we have assessed the overall lipid retention rate of GAMSII samples with and without such a surfactant treatment (**Fig. S1**). Briefly, we prepared four samples (all treated with AcX for protein anchoring) with different sample homogenization/digestion conditions: (1) proK with surfactants,

(2) proK only (no surfactants), (3) trypsin with surfactants, and (4) trypsin only (no surfactants). As result, we found that removing the surfactants from the digestion buffer drastically improved the overall lipid retention. We also observed that the use of weak proteolysis (= trypsin) led to a higher retention rate. This could be due to the fact that peptide fragments from the proK digestion are much smaller than those after trypsin digestion and some of the lipid-protein hydrophobic interaction sites are lost for the proK digestion.

When combined, these results and analyses suggest that non-covalent interactions, such as hydrophobic interactions, could indeed mediate endogenous lipid retention in GAMS. While direct covalent anchoring of lipids to the hydrogel is possible (see our response to Reviewer 1), we believe the non-covalently bound lipids play a more dominant role in the subsequent MALDI desorption and ionization step. In summary, we have included a detailed discussion on the chemical basis of GAMS (Page 3, Line 88-102) as well as the aforementioned surfactant treatment results (**Fig. S1**) in the revised manuscript.

2. The workflow presented in Figure 1 is somewhat misleading as it fails to include the crucial step of antibody labeling for targeted proteome imaging, even though the main text mentions that photocleavage occurs on the antibodies conjugated with Photocleavable mass-tag (PC-MT) modification. Due to the lack of this step, it is unclear which step is responsible for lipid releasing and which one for antibody releasing. Therefore, Figure 1 should be updated to include the antibody labeling step and the magnified insets should display the antibodies to avoid any confusion.

Following the Reviewer's comment, we have created two separate figures to illustrate the generalized vs. specific (e.g., the sequential lipid-protein imaging pipeline with the PC-MT modification) GAMS workflow to avoid potential confusion. Specifically, we have updated **Fig. 1** to illustrate the generalized GAMS workflow and added a new figure (**Fig. S13**) to illustrate the sequential GAMS workflow of lipid imaging + PC-MT modification/protein imaging.

3. The content in lines 62-68 of the results section needs to be relocated to the introduction section for better clarity. Ref 19 is the most relevant reference to this work because it also combines MSI and expansion microscopy for high resolution. It should be described in detail in the introduction.

Following the Reviewer's suggestion, we have moved the contents in lines 62-68 of the previous results section (which includes Ref 19) to the introduction (Page 2, Line 52-66). We have also included in the same paragraph a detailed description of Ref 19 (now Ref 25) as well as other literature to better lay out the methodological landscape of the field.

4. The authors should provide a more detailed introduction on the working principle of expansion microscopy and its various applications in proteomics, lipidomics, and multiomics.

Following the Reviewer's suggestion, we have provided additional descriptions on the working principle of expansion microscopy as well as its broad applications in spatial multi-omics and fluorescence imaging in the introduction (Page 2, Line 55-61).

5. In line 74, it is unclear how "the target biomolecules can be retained via physical interaction with the hydrogel itself". Further explanation or an example is needed.

We agree with the Reviewer that our description of "physical interaction with the hydrogel itself" was confusing and requires additional explanation. In response, we have removed this description from the sentence and instead focused on "the physical interaction with other molecules covalently anchored to the hydrogel polymer chains". This is because for the latter we have a clear example to provide (= non-covalent retention of the lipids; see our discussion in Q#1). We have included this example and associated discussion in the revised main text (Page 3, Line 88-102).

6. In line 77, explain how "the target biomolecules can be chemically tethered to the hydrogel polymer chains using a cleavable small molecule linker".

In response, we have included a detailed description of how antibodies possessing a photocleavable mass reporter can be covalently anchored to the hydrogel polymer network (and subsequently cleaved to release the mass reporter) as an example of "target biomolecules...tethered to the hydrogel polymer chains using a cleavable small molecule linker" in the revised main texts (Page 3, Line 103-110).

7. The dynamic range of the lipidome detection should be discussed. How many different types of lipids are detected? How different is the GAMS I lipidomics from the standard non-expansion MSI?

To address the Reviewer's comment, we have performed in-depth analyses of the MS profile of fresh-frozen and GAMS I sample, and added related discussions in the revised main text (Page 4, Line 154-168; Page 5, Line 189-199). Briefly, we have created a Venn diagram (**Fig. S6a, Fig. S12a**) to illustrate the common vs different lipid peaks detected in the two samples. We found that ~50-60% of the fresh-frozen peaks were captured by the GAMS I sample and this percentage increased as we switched to a MALDI-TOF mass spectrometer with higher sensitivity. To evaluate GAMS I lipidomics and its difference from the standard non-expanded sample, we further analyzed the chemical composition (= lipid types) of lipid peaks detected in the fresh-frozen vs. GAMS I sample (**Fig. S6b, Fig. S12b**). As result, we found that the lipid composition of fresh-frozen vs. common vs. GAMS I lipid peaks remained largely consistent, suggesting that the chemical perturbation of GAMS I is minimal. Interestingly, the ratio of phospholipids possessing a primary amine head group (= PE + PS) also remained consistent across the fresh-frozen (~48%) vs. common (~48%) vs. GAMS I (~49%) lipid peaks. As described previously, the GAMS I-processed sample goes through additional chemical processes, such as the chemical fixation (e.g., with PFA) and NHS-based membrane protein anchoring step. In fact, both of these steps can covalently anchor PE and PS to other molecules

or to the hydrogel network and hence can hinder their ionization in the subsequent steps. The fact that PE and PS peaks were not specifically suppressed by GAMSI suggests that a majority of these phospholipids were not as chemically affected (= modified by the fixation or the AcX anchoring) as we speculated. Combined, these analyses provide preliminary yet critical insights into the chemical perturbation of GAMSI, which we have included in the revised main text (Page 4, Line 154-168; Page 5, Line 189-199).

8. In the method section, provide the catalog numbers of the PC-MTs. If they are customized orders, provide sequences of the MTs. Provide the chemical name of the PC linker, eg. nitrobenzyl. Briefly describe the antibody conjugation method. Catalog numbers of all other chemical and biochemical reagents should be included.

Following the Reviewer's suggestion, we have included the catalog number, the mass reporter m/z value, and the vendor's name (= AmberGen, Inc) for all the PC-MTs we used. While the general chemical design of the PC linker is well known for PC-MT (= *o*-nitrobenzyl ester; we included its chemical structure in the new figure **Fig. S13**), specific conjugation/photocleaving chemistries of the commercial PC-MTs are proprietary and are not disclosed to the users. Therefore, we hope we could defer the details of PC-MT conjugation/photocleaving chemistries to the vendor by providing their full catalog numbers. Finally, we have checked the description of other reagents in our method section and added catalog numbers and vendor names (especially for those purchased from uncommon vendors).

Reviewer #3:

This paper describes an imaging workflow to interrogate molecular species using mass spectroscopy on samples that have been expanded in size by coupling the tissue to an absorbant hydrogel that swells in size. The goal is to get sub-micrometer resolution to overcome the spatial limitations of point scanning in mass spectroscopy. Once this approach reaches subcellular resolution it may be quite useful in multiplexing many molecular labels on the same tissue sample, a potentially valuable approach to add to the armamentarium that presently relies heavily on the very powerful approaches of spatial transcriptomics.

We thank the Reviewer for commenting on the significance as well as potential impact of our work for single-cell/subcellular spatial omics.

The main problem with this paper is that the promise of sub-micrometer resolved images is not fully realized. None of the images show cellular, much less subcellular resolution. Fig 5b shows phospholipids as white, but a blow-up of that image shows phosphatidic acid (according to the label in the figure) as black- but the images look the same, so something is amiss. In either case however, there are no clear cellular details. The granule cell layer which is densely packed with neurons and neuropil so there is little to see that would require submicron resolution. Perhaps the Purkinje cell layer which has substantially larger cells and cells that are in a single line would

be a better target for this proof of concept. The arrow heads may be pointing at some cell type as suggested by the authors, but the fluorescence image provided as evidence is grossly subpar. The two channels should be shown separately. I would have expected to see some single myelinated axons in the granule cell layer- given the submicron resolution but none were apparent.

We thank the Reviewer for carefully reading through our manuscript and providing valuable comments on the weaknesses of our manuscript and how we could improve. Following the Reviewer's suggestion, we have performed additional lipid GMSI experiment on the Purkinje cell layer to demonstrate the (sub)cellular imaging capabilities of our approach. As result, we successfully resolved individual Purkinje cells located between the granule cell layer and the molecular layer in the mouse cerebellum. The observed enrichment of $m/z = 715.65$ lipid peak was consistent with the previously reported presence of sphingomyelin SM(d18:1/18:0) in the mouse anterior cerebellum and its localization within the Purkinje cell layer (with lower imaging spatial resolution) (see Blot et al., *PNAS* **118**, e2016969118 (2021) and Sugimoto et al., *Biochimica et Biophysica Acta* **1851**, 1554-1565 (2015)). We have included these new imaging results as **Fig. 5a** to replace some of the old **Fig. 5b and 5c** figure panels (which did not fully demonstrate the resolving power of GMSI as the Reviewer pointed out). As a cross-validation of the location and the identity of the Purkinje cells, we have also provided in **Fig. 5a** a high-quality fluorescence image of the same tissue section with the Purkinje cells fluorescently labeled.

Lastly the power of this approach requires showing a high degree of multiplexing. Two labels is not sufficient and showing more would be more compelling.

Following the Reviewer's suggestion, we performed additional rounds of GMSI experiments with more than two protein labels (= the photocleavable mass-tag conjugated antibodies) to demonstrate our approach's multiplexing capability. In principle, if the mass-tag conjugated antibodies label our tissue well, there shouldn't be a limit to the number of proteins we can image concurrently, because all the antibodies are primary and won't have cross-talks in between.

However, while we were able to scale up the imaging to more than two labels, we observed some technical issues with the additional mass-tag conjugated antibodies we used (see the multiplexed imaging results below). For example, when we applied the NeuN (a nuclear maker for neurons; AP1001201, AmberGen, Inc.; $m/z = 1308.71$) and parvalbumin (PVALB; a cytosolic marker for inhibitory interneurons; AP1001203, AmberGen, Inc.; $m/z = 1539.79$) probes to the GMSI sample, we observed either a severe aggregation of the probe or low signals counts. Given that the probe production may have batch-to-batch difference, we repeated the same staining/imaging experiment with three separate batches, but ended up encountering the same issue. Following these results, we presumed that the conjugated antibody itself could be the issue, provided that the mass-tags should work robustly across all the probes from their product specification. Because the customers/users do not have control over which antibodies are conjugated to the mass-tags (otherwise it would require custom conjugation, which we could not

afford), we were not able to request switching the antibodies to those we validated would work with our sample. Nevertheless, we included the multiplexed GMASI images from these imaging experiments below. To further improve the image quality, we recognized it is critical to validate every mass-tag conjugated antibody and to confirm its compatibility with each tissue type and sample preparation protocol. For example, the customers/users may want to test the antibody staining quality first with immunofluorescence and then request a custom conjugation with a mass-tag (which, however, requires additional costs) to enhance the staining/imaging quality. To recapitulate these experiments and results, we have also included an in-depth discussion on multiplexed protein labeling in the main text (Page 6, Line 212-217).

Spatial distribution of photocleavable mass-tag conjugated antibodies targeting MBP, SYN-I, and NeuN in a ~4-fold expanded mouse cerebellum lobule (PFA-fixed). We observed a low signal-to-noise (< 3:1) ratio for the NeuN probe. Scale bars: 200 μ m (760 μ m).

Spatial distribution of photocleavable mass-tag conjugated antibodies targeting MBP, SYN-I, and PVALB in a ~4-fold expanded mouse cerebellum lobule (PFA-fixed). We observed severe aggregation for the PVALB probe. Scale bars: 200 μ m (760 μ m).

In sum the technique needs to resolve cellular or subcellular details to be of value in brain samples and multiple protein labels (>2) at the same time would also be helpful.

Again, we are grateful to the Reviewer's suggestions on how to improve our technique and manuscript. In summary, we have successfully demonstrated imaging of (sub)cellular details by focusing on the Purkinje cell layer as suggested by the Reviewer. We have also performed multiplexed imaging of more than two protein labels using additional mass-tag conjugated antibodies procured from the vendor. We think the final staining/imaging quality can be further improved in future studies by addressing the technical issues we encountered with antibody aggregation and low staining quality.

Reviewers' Comments:

Reviewer #1:

Remarks to the Author:

The authors have made significant and well justified revisions to the manuscript which address most of my points satisfactorily.

I appreciate the clarification in terminology and figures which will assist non-experts to better follow the work.

I also appreciate the more detailed analysis of the lipid profiles via molecular annotation to better understand the nature of the detected species and differences between workflows. This is a really important part of the story and key for validating the methodology.

I have one minor change I recommend for the main text, on Line 185 the authors state "any MALDI-TOF mass spectrometer". Here, the inclusion of "TOF" is not necessary as the approach should be applicable to any MALDI mass spectrometer, not just TOF instruments (e.g. also FTMS).

Overall I am convinced that this approach is valuable to the community, that the work has been sufficiently validated for this stage, and that the manuscript is of high enough quality for publication.

Reviewer #2:

Remarks to the Author:

The authors addressed my questions and concerns.

Reviewer #4:

Remarks to the Author:

In this manuscript the authors describe an optimization method for the spatial resolution of mass spectrometry imaging (MSI), using gel-assisted expansion. Gel-Assisted Mass Spectrometry Imaging (GAMSI) pushes the limits of spatial resolution of mass spectrometry by enhancing the volume up to 6-fold, similar to fluorophore-based expansion microscopy.

The manuscript provides a detailed description of the novel approach and a thorough validation of the effects of sample expansion, using the cerebellum to test the spatial resolution. The cerebellum is a brain region with a highly organized, crystalline structure, ideal to test potential distortions due to the expansion. Overall, I think the manuscript is of high quality. The novel approach is well-described, validation experiments are largely appropriate and the work forms an important step forward.

I do have one concern, related to the experiments in which the spectrometry is combined with fluorescence imaging and the conclusions that are made based (partially) on those results. In figure 5, the authors aim to compare the GAMSI images to a (standard) immunostainings of Purkinje cells against calbindin. The GAMSI image does indeed appear to label Purkinje cells, as suggested by the calbindin immuno. However, the quality of the calbindin staining image is of such poor quality that it is difficult to assess how accurate the GAMSI image is. Is that the result of the processing? With this image quality I would be very hesitant to claim with certainty that the lipids indeed colocalize with Purkinje cells, even though it is very likely. The quality definitely affects the ability to make conclusions about the subcellular elements, a term that is used by the authors in eg. the abstract. Without a comparison to a sharper image, I think any claims about subcellular or (sub)cellular features should be avoided. I would encourage the authors to try to optimize immunostaining image quality, as (reasonably) sharp immunos would greatly improve the power to interpret the GAMSI outcomes at the cellular, and perhaps even subcellular, level.

We thank the reviewers for carefully reviewing our revised manuscript and for providing additional feedback to improve its quality. To address Reviewer 4's concern, we have elaborated on the possible cause of the low immunostaining quality (due to the standard tissue format used in mass spectrometry imaging), performed additional validations of the Purkinje cell/lipid peak colocalization, and removed all the subcellular claims as suggested by the Reviewer. We hope this updated version of our manuscript is acceptable to Nature Communications.

Reviewer #1:

The authors have made significant and well justified revisions to the manuscript which address most of my points satisfactorily. I appreciate the clarification in terminology and figures which will assist non-experts to better follow the work. I also appreciate the more detailed analysis of the lipid profiles via molecular annotation to better understand the nature of the detected species and differences between workflows. This is a really important part of the story and key for validating the methodology.

I have one minor change I recommend for the main text, on Line 185 the authors state "any MALDI-TOF mass spectrometer". Here, the inclusion of "TOF" is not necessary as the approach should be applicable to any MALDI mass spectrometer, not just TOF instruments (e.g. also FTMS).

We thank the Reviewer for pointing this out. Following the Reviewer's suggestion, we removed "TOF" from the main text so that the sentence reads "... any MALDI mass spectrometer ..." (Page 5, Line 184-185).

Overall I am convinced that this approach is valuable to the community, that the work has been sufficiently validated for this stage, and that the manuscript is of high enough quality for publication.

Once again, we are grateful to the Reviewer for reading through our revised manuscript and for highlighting the quality and significance of our work.

Reviewer #2:

The authors addressed my questions and concerns.

We thank the Reviewer for all the valuable comments and suggestions that helped improve our manuscript.

Reviewer #4:

In this manuscript the authors describe an optimization method for the spatial resolution of mass spectrometry imaging (MSI), using gel-assisted expansion. Gel-Assisted Mass Spectrometry Imaging (GAMSI) pushes the limits of spatial resolution of mass spectrometry by enhancing the volume up to 6-fold, similar to fluorophore-based expansion microscopy.

The manuscript provides a detailed description of the novel approach and a thorough validation of the effects of sample expansion, using the cerebellum to test the spatial resolution. The cerebellum is a brain region with a highly organized, crystalline structure, ideal to test potential distortions due to the expansion. Overall, I think the manuscript is of high quality. The novel approach is well-described, validation experiments are largely appropriate and the work forms an important step forward.

We are grateful to the Reviewer for carefully reading through our revised manuscript and for highlighting the high quality and importance of our work. We also thank the Reviewer for kindly noting the appropriateness of the validation experiments we have performed.

I do have one concern, related to the experiments in which the spectrometry is combined with fluorescence imaging and the conclusions that are made based (partially) on those results. In figure 5, the authors aim to compare the GAMSI images to a (standard) immunostainings of Purkinje cells against calbindin. The GAMSI image does indeed appear to label Purkinje cells, as suggested by the calbindin immuno. However, the quality of the calbindin staining image is of such poor quality that it is difficult to assess how accurate the GAMSI image is. Is that the result of the processing? With this image quality I would be very hesitant to claim with certainty that the lipids indeed colocalize with Purkinje cells, even though it is very likely. The quality definitely affects the ability to make conclusions about the subcellular elements, a term that is used by the authors in eg. the abstract. Without a comparison to a sharper image, I think any claims about subcellular or (sub)cellular features should be avoided. I would encourage the authors to try to optimize immunostaining image quality, as (reasonably) sharp immunos would greatly improve the power to interpret the GAMSI outcomes at the cellular, and perhaps even subcellular, level.

First, we would like to thank the Reviewer for the detailed comments and suggestions on the immunostaining. As the reviewer pointed out, while we were able to identify Purkinje cell bodies using the calbindin staining in Fig. 5a, a good portion of their subcellular features, such as their branched dendritic trees, were lost due to the low intensity and heightened background.

To address the Reviewer's question, we believe that the observed low immunostaining quality was likely caused by the tissue processing format used for MALDI mass spectrometry imaging (MALDI-MSI). Currently, fresh-frozen tissue is arguably the standard format for MALDI-MSI sample preparation and is widely accepted by the field. This is because such preparation would better preserve the spatial organization of lipids and metabolites native to the tissue and provide better extraction/ionization efficiency in the final MSI step with little to no chemical crosslinking by the fixatives. However, during the tissue freezing, storage, and thawing steps (which are often performed at dry ice temperature or above), a large portion of the protein 3D/2D epitopes

and tissue ultrastructures are lost due to the formation of water crystals. This can then lead to reduced specificity of antibodies as well as enhanced unspecific binding.

To validate that the observed immunostaining quality is indeed caused by MALDI-MSI tissue preparation (and not by our immunostaining protocol or the subsequent sample gelation and expansion process), we sacrificed a new animal and prepared perfusion-fixed (4% PFA, following the standard tissue preparation format for immunohistochemistry) mouse cerebellum slices of the same thickness as its fresh-frozen counterpart. We then applied the same calbindin antibodies (and the same secondary antibodies) to both the fresh-frozen (as control) and the perfusion-fixed tissue slices. As a result, while we were able to identify Purkinje cell bodies in both tissue types, the perfusion-fixed tissue showed much improved immunostaining quality and enriched subcellular details. This validates our hypothesis that the fresh-frozen tissue preparation format preferred by MALDI-MSI is a major factor leading to the lower immunostaining quality.

Calbindin immunostaining of fresh-frozen (left) and perfusion-fixed (right) ~25 μm thick mouse cerebellum slices. The same calbindin antibodies and immunostaining protocols were used. Scale bars: 500 μm (top row) and 20 μm (bottom row). ML: molecular layer; GL: granular cell layer.

Following this result, ideally, we would like to switch from fresh-frozen to perfusion-fixed tissue for better immunostaining quality. However, when we tested GAMSIs on the perfusion-fixed brain slice, we observed no lipid signals on the mass spectrometer (rapifleX). We suspect this is due to the denser chemical crosslinking and lower analyte extraction/ionization efficiency of the perfusion-fixed tissue, as described earlier. Since perfusion-fixed tissue remains incompatible with GAMSIs in our hands, we had no choice but to revalidate the observed colocalization using

fresh-frozen brain slices. The next figure shows the lipid GAMSIs and calbindin staining results of an additional fresh-frozen sample that we tested (along with the results from Fig. 5a for comparison). While we could not significantly improve the immunostaining quality as the tissue format remained fresh-frozen, we observed similar enrichment of GAMSIs lipid peaks in the Purkinje cell bodies. It is worth noting that not all Purkinje cells show enrichment of this lipid peak, which we believe may reflect the different metabolic states of the Purkinje cells. We hope these results can serve as additional validation for the Purkinje cell/lipid peak colocalization observed in Fig. 5a and help reduce the Reviewer's concerns.

Following the above-mentioned trade-offs of MALDI-MSI quality vs. immunostaining quality, we note that a different tissue fixation strategy may need to be developed and validated for MALDI-MSI in future studies, ideally with a chemical strength in between the fresh-frozen and perfusion-fixation (e.g., high-pressure freezing + freeze substitution?). Optimizing such fixation protocols and validating their metabolic/proteomic profiles will take substantial time and effort but will be informative to the community. However, whether such protocols can achieve both high-quality MALDI-MSI (GAMSIs) and immunostaining still remains unclear at this point.

Fluorescence image (left, calbindin immunostaining), lipid GAMSIs image (middle, $m/z = 715.65$), and overlay image (right) of ~ 4 -fold expanded fresh-frozen mouse cerebellum slices (prepared from two

different animals). The mass spectrometry instrument (Bruker rapifleX) pixel size is set at 10 μm . Scale bars: 125 μm (500 μm post-expansion, top row of each sample) and 25 μm (100 μm post-expansion, bottom row of each sample). ML: molecular layer; GL: granular cell layer; WM: white matter.

Finally, we understand the Reviewer's remaining concerns about our claims and usages of "subcellular" or "(sub)cellular" features in the manuscript. In its strict sense, we agree that we have not fully demonstrated GAMSIs of subcellular features (e.g., GAMSIs of single neurites or single organelles) given the above discussed challenges of tissue format preferred for mass spectrometry imaging. Therefore, to follow the Reviewer's suggestion, we have removed or changed all our descriptions of "subcellular" or "(sub)cellular" features to "cellular" in the abstract and main text (see highlighted texts, e.g., Page 1, Line 27; except for places we described the capability of existing MSI methods). We look forward to future developments in our group and in the community to demonstrate subcellular GAMSIs.

Reviewers' Comments:

Reviewer #4:

Remarks to the Author:

I want to thank the authors for the detailed explanation for the cause of the issue I raised, and all the efforts they made to solve it. Together with the corrections to the text, the response sufficiently addresses the issue. The current presented data support the claim the imaging quality is high enough to perform analyses at the cellular level.